# High-resolution mapping of on-road vehicle emissions with real-time traffic datasets based on big data

Yujia Wang[1], Hongbin Wang[2], Bo Zhang[2], Peng Liu[2], Xinfeng Wang[1,*], Shuchun Si[3], Likun Xue[1], Qingzhu Zhang[1], Qiao Wang[1]

[1]Environmental Research Institute, Shandong University, Qingdao 266237, China

[2]Traffic Police Detachment of Jinan Public Security Bureau, Jinan 250014, China

[3]School of Physics, Shandong University, Jinan 250100, China

*Correspondence to: Xinfeng Wang (xinfengwang@sdu.edu.cn)

**Abstract.** On-road vehicle emissions play a crucial role in affecting fine-scale air quality and exposure equity in traffic-dense urban areas. They vary largely in both spatial and temporal scales due to the complex distribution patterns of vehicle types and traffic conditions. With the deployment of traffic cameras and big data approaches, we established a bottom-up model that employed interpolation to obtain a spatially continuous on-road vehicle emission mapping for the main urban area of Jinan, revealing fine-scale gradients and emission hotspots intuitively. The results show that the hourly average emissions of nitrogen oxides, carbon monoxide, hydrocarbons, and fine particulate matters from on-road vehicles in urban Jinan were 345.2, 789.7, 69.5, and 5.4 kg, respectively. The emission intensity varied largely with a factor of up to 3 within 1 km on the same road segment. The unique patterns of road vehicle emissions within urban area were further examined through time series clustering and hotspot analysis. When spatial hotspots coincided with peak hours, emissions were significantly enhanced, making them key targets for traffic pollution control. Based on the established emission model, we predicted that the benefits of vehicle electrification in reducing vehicle emissions could reach 40% – 80%. Overall, this work provides new methods for developing a high-resolution vehicle emission inventory in urban areas, and offers detailed and accurate emission data and fine spatiotemporal variation patterns in urban Jinan, which are of great implications for air pollution control, traffic management, policy making, and public awareness enhancement.

## 1. Introduction

The rapid increase in the number of vehicles in recent years has brought convenience to people's travel and daily lives. At the same time, it has also posed considerable challenges, including traffic congestion, severe air pollution, and adverse health impacts (Uherek et al., 2010; Guo et al., 2014; Zhang et al., 2018; Shi et al., 2023). Particularly, in urban areas with dense populations, vehicle emission acts as a major cause of air quality deterioration, which has attracted widespread public attentions

(Ramacher et al., 2020; Qi et al., 2023). To evaluate the environmental impacts, researchers and stakeholders have established on-road vehicle emission inventories to estimate the amount of air pollutants emitted from vehicle exhausts into the atmosphere

(Zhang et al., 2018). With consideration of vehicle population growth, stringent emission standards, the phasing out of old vehicles, and vehicle electrification, the spatiotemporal emission characteristics of vehicles in new period may have changed (Wen et al., 2023; Zhu et al., 2023). Therefore, developing up-to-date dynamic on-road vehicle emission inventories that align with the current urban traffic flow conditions is urgently needed and is quite important for vehicle pollution control and urban air quality improvement.

Emission factors, defined as the number (or mass) of pollutants emitted per unit of activity, are one of the basic data for the development of emission inventories. Several methods for determining on-road vehicle emission factors have been used in real-world conditions, such as street canyon and road tunnel studies (Brimblecombe et al., 2015; Zhang et al., 2015), remote sensing (Davison et al., 2020), and on-board measurements (Jaikumar et al., 2017). Other methods involve the estimation of air dispersion by tracking a trace gas that can be added or emitted by traffic (Belalcazar et al., 2010) and the use of the eddy-

covariance (EC) technique to quantify emissions based on atmospheric turbulence (Conte and Contini, 2019). Although previous measurements have yielded convincing emission factors for local vehicles, some limitations in the development of emission inventories have not yet been adequately addressed. Road traffic sources, belonging to mobile sources, are characterized by low emission heights, densely populated emission areas, and obvious spatiotemporal heterogeneity (Liu et al., 2018; Ding et al., 2023). The traffic flows, vehicle compositions, and vehicle speeds vary dramatically over short periods

and distances, affecting the emission characteristics of traffic sources (Chen et al., 2020; Jiang et al., 2021). As a result, the accuracy of vehicle emission inventory largely depends on the quality of the input data on traffic conditions (Ding et al., 2021; Romero et al., 2020). Conventional vehicle emission inventories were usually established by using a top-down approach, based on statistical data including vehicle populations, mileages, fuel types, etc (Cai and Xie, 2007; Fameli and Assimakopoulos, 2015). These inventories are generally temporally static and spatially rough, lacking high-spatiotemporal-resolution data to

characterize road vehicle emission. In the recent dozen of years, several advanced technologies such as GPS-equipped floating cars, open-access congestion maps, and intelligent transportation system (ITS), have been applied to develop high-resolution traffic emission inventories (He et al., 2016; Zhang et al., 2018; Liu et al., 2018; Maes et al., 2019; Ghaffarpasand et al., 2020). Exploiting taxi GPS data can infer the spatial and temporal variation patterns of urban traffic emissions and the heaviest traffic volumes and emission hotspots were often discovered in the city center (Luo et al., 2017; Liu et al., 2019). Open congestion

maps provide real-time traffic information, making traffic volume inference timely. A more detailed vehicle emission inventory in Beijing was developed based on open congestion maps, indicating significant impacts on vehicle emissions caused by the traffic restrictions (Yang et al., 2019). In addition, several studies have substantially improved vehicle emission inventories by collecting detailed, high-precision, and real-time monitoring data with the ITS. For example, the spatial resolution of hourly emission inventory in Xiaoshan District in Hangzhou was increased by 1–3 orders of magnitude through

ITS (Jiang et al., 2021). In their study, emission hotspots exhibited sharp small-scale variability and would strengthen during peak hours. By means of ITS, the spatiotemporal dynamics of vehicle emissions in Guangdong Province were also revealed

(Ding et al., 2023). It was found that gasoline passenger cars were the main contributors to carbon monoxide (CO) and hydrocarbons (HC) emissions during peak hours, while diesel trucks were the dominant source for the emissions of nitrogen oxides ($NO_x$) and fine particulate matters ($PM_{2.5}$) at night. Though above improvements, the previous inventory compilation

techniques for on-road vehicles have some limitations due to incomplete traffic data, insufficient vehicle details, or high costs for wide coverage. More comprehensive data needs to be obtained in cost-effective methods to achieve long-term and wide-area coverage, particularly in less developed cities.

Recently, traffic camera networks have been widely adopted and readily available, providing extensive coverage in critical urban areas. They have the capability to continuously capture nearly all vehicles driving on roads (Liu et al., 2024),

thus providing an opportunity to develop an ultra-fine resolution vehicle emission inventory. To some extent, large-scale real-time traffic datasets are crucial for elucidating the spatiotemporal variations in on-road vehicle emissions (Wu et al., 2022). However, the main concern is that processing and analyzing a large amount of data are challenging and time-consuming tasks (Lv et al., 2023). Compared to traditional statistical methods, big data approaches offer apparent advantages in handling, validating, analyzing, mining, and visualizing large-scale, multi-source, and structurally complex monitoring data. At present,

big data has been used for air pollution mapping with much higher spatial precision than fixed-site monitoring (Apte et al., 2017). In addition, Deng et al. (2020) used a big data approach to establish a high-resolution and large-region vehicle emission inventory in the Beijing–Tianjin–Hebei region, but only for trucks. With the application of big data techniques, it becomes feasible to develop accurate, practical, and dynamically updatable urban traffic emission inventories with high spatiotemporal resolution.

Note that even widely distributed traffic cameras are unlikely to achieve spatially continuous observations. The spatial gaps of meters to kilometers between fixed traffic cameras determine the upper resolution limit of the bottom-up on-road vehicle emission inventory. Therefore, alternative approaches are needed to complement the observation data and fill the gaps. Interpolation, a useful data processing method, can preserve the emission data at the original monitoring points while maximizing the reproduction of the spatial gradient of emissions over short distances (10 m–1 km). Jeong et al. (2019)

innovatively applied spatial interpolation models to achieve a more accurate estimation of methane emission from a landfill. Similarly, exploiting spatial interpolation to the estimation of vehicle emissions can compensate for spatial gaps between traffic monitoring points, making on-road vehicle emission mapping dynamic and continuous. Due to the drastic change in air pollutant concentrations over short distances in urban areas caused by the uneven spatial distribution of traffic sources (Apte and Manchanda, 2024), refined vehicle emissions through interpolation will of course provide an effective reference for the

improvement and interpretation of air pollution mapping.

As the capital of Shandong Province and an important transportation hub in northern China, Jinan is confronting severe air pollution. The number of vehicles in Jinan has exceeded 3 million in 2020, and vehicular emissions are a significant source of urban air pollutants. Jinan has basically achieved full coverage of the traffic camera monitoring, which allows us to obtain traffic data for each road segment. In this study, we combined traffic camera recordings with field surveys, making the

95 framework applicable to most cities. By using a bottom-up approach to calculate emissions and pioneering the application of

spatial interpolation, we successfully established a high-resolution (temporal resolution of 1 h and spatial resolution of 50 m × 50 m) on-road vehicle emission mapping. With the high-efficiency processing capabilities of the big data approaches and the accumulation of hourly data for nearly one year, the seasonal variations, weekday and weekend differences, and diurnal changes as well as the spatial distribution patterns of vehicle emissions for multiple pollutants were analyzed and revealed.

Through time series clustering and hotspot analysis, the different diurnal variation patterns and hotspot areas of vehicle emissions on urban roads were further identified. Additionally, considering the rapid development of new energy vehicles (NEVs), future scenarios were designed to predict the positive impact of vehicle electrification on on-road traffic emissions. Finally, the validation of the developed on-road vehicle emission inventory was conducted by comparing with other inventories.

## 2. Methodology and data

### 2.1 Road network and real-time traffic monitoring

Jinan is located in the middle of the Beijing-Tianjin-Hebei region and the Yangtze River Delta region, serving as an important urban transportation hub in northern China. As of 2023, it had a population of over 9.43 million, with a GDP of CNY 1276 billion. The total length of the road network was around 18356 km within a geographical area of 10244 km$^2$. There were 3.39 million private vehicles (motorcycles excluded) in Jinan, with an average annual growth rate of 7% since 2019. However, the

110 construction of roads and rail transit lagged behind relatively. Limited traffic space was incompatible with the rapidly increasing number of vehicles, leading to frequent traffic congestion, especially in urban areas. In addition, driving restrictions for trucks in peak hours were implemented in the main urban area by the local government. In this study, the main urban area of Jinan (within the Inner Ring Expressway) was selected to collect real-time traffic data from monitoring points and further calculate the on-road vehicle emissions (see Fig. 1). There were total 1189 traffic monitoring points in the main urban area,

which captured all vehicles passing by with fast cameras, identified the vehicle categories, and recorded the traffic flows. Four categories of vehicles were recognized automatically, including light-duty vehicles (LDVs), heavy-duty vehicles (HDVs), new energy light-duty vehicles (NELDVs), and new energy heavy-duty vehicles (NEHDVs). All the roads in the main urban area were classed as highways, expressways, arterial roads, or minor arterial roads and were divided into numerous segments by traffic monitoring points. The gaps between two monitoring points ranged from 10 m to 3 km. The hourly data of vehicle flows

and categories were obtained with image processing, object detection, and image recognition algorithms.

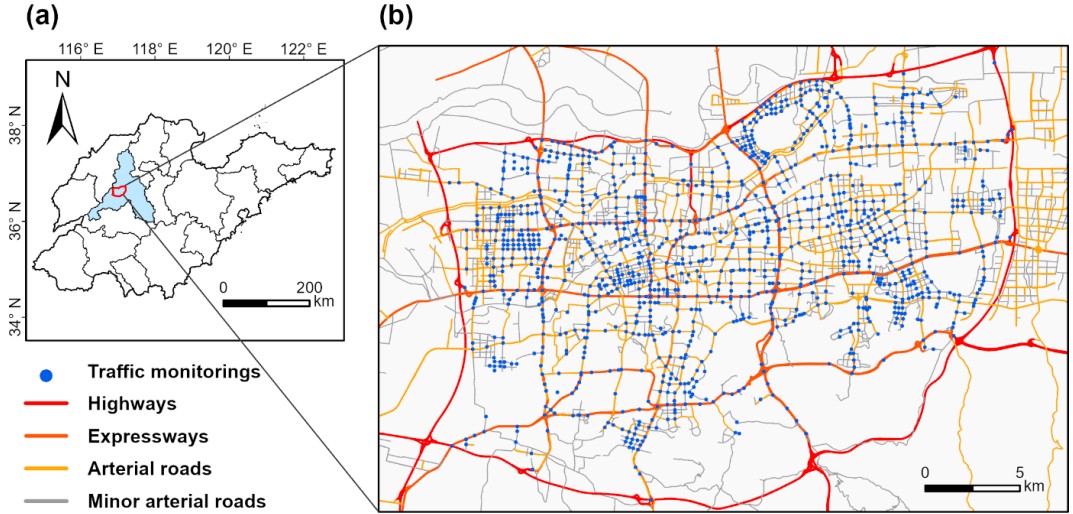

**Figure 1.** Road network and the real-time traffic monitoring points in the main urban area of Jinan. (a) Map shows Jinan City (the blue area) located in Shandong Province, China, with the main urban area within the red borders. (b) Real-time traffic monitoring achieved full coverage over the main urban area of Jinan.

## 2.2 Data collection and processing based on big data approaches

Big data methods were employed in this study to calculate the high-resolution emissions of air pollutants. The accuracy largely depended on the quality of input data of traffic and meteorological conditions (Romero et al., 2020). To obtain real-time traffic information, the hourly flow data for four categories of vehicles measured at each monitoring point were collected from 1 April 2023, to 29 February 2024. The fractions of specific vehicle types were determined based on our previous surveys conducted on typical roads within urban areas of Jinan in April, 2022 (Wang et al., 2024), and the classification method was introduced in Section 2.3. The hourly meteorological data during the study period at the surface level were adopted from ERA5 (https://cds.climate.copernicus.eu) with a horizontal resolution of 0.25 degrees (Hersbach, et al., 2023). They were integrated with the traffic dataset with higher resolution by a "snapping" procedure on the basis of nearest geographical coordinates. Meteorological data were used for environmental corrections of emission factors. In this process, we determined the local temperature and humidity ranges rather than the exact values, and assigned specific correction coefficients accordingly.

With the hourly traffic and meteorological data, the on-road vehicle emissions of primary pollutants, including $NO_x$, CO, HC, and $PM_{2.5}$ were calculated and visualized. Figure 2 shows the framework of data processing for vehicle emission calculation and mapping. Specifically, the original dataset of traffic monitoring with approximately 106 million records were collected and integrated with meteorological data for subsequent data storage and management. Then, the data underwent extensive pre-processing including cleaning, integration, transformation, and reduction. After that, the original data was consolidated into a structured dataset comprising 5.5 million sets of records. Each set of record included multiple parameters of traffic flows for different vehicle types, emission correction factors (depending on meteorological conditions and vehicle speed), road information, and timestamps. They were further used to calculate vehicular emissions and analyze the

spatiotemporal distribution characteristics, ultimately aiding to understand the status of urban vehicle emissions and

145 formulating corresponding control strategies.

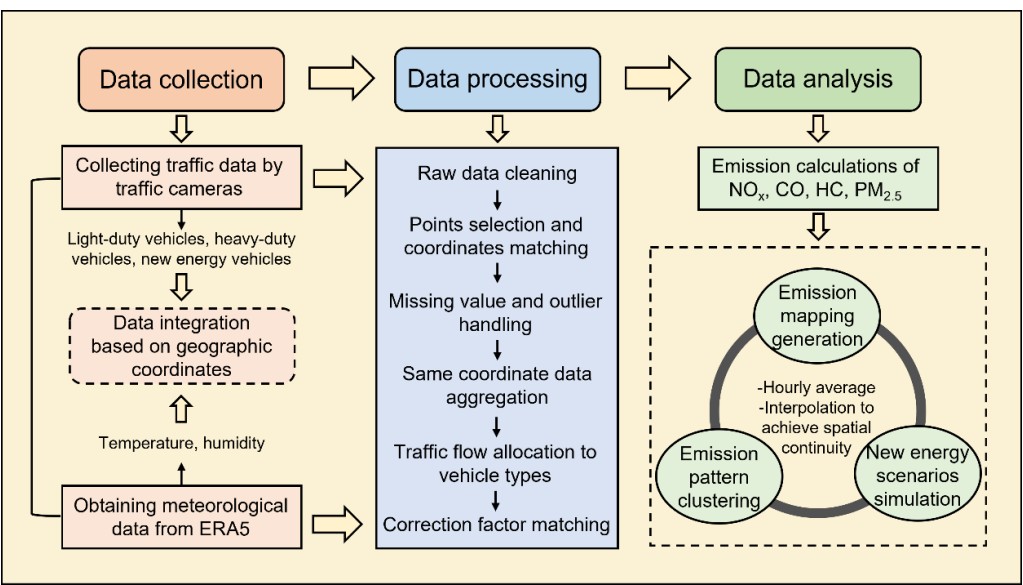

**Figure 2.** Model framework for high-resolution mapping of on-road vehicle emissions based on big data approaches.

### 2.3 Vehicle emission calculations and hyperfine-resolution mapping

The hourly emissions of air pollutants including $NO_x$, CO, HC, and $PM_{2.5}$ at each road segment were calculated based on traffic

flows, vehicle speeds, vehicle categories, road segment length and emission factors (see Eq. 1) (Zhang et al., 2016; Yang et al., 2019; Jiang et al., 2021). With consideration of emission variations caused by local conditions, localized correction coefficients were adopted in this study for adjustment in emission factors (see Eq. 2).

$$E_{h,l,j} = \sum_t EF_{c,j} \times TF_{c,h,l} \times L_l, \tag{1}$$

Where, $E_{h,j,l}$ is the total emission of pollutant $j$ on road link $l$ at hour $h$, in units of grams per hour (g h$^{-1}$). $EF_{c,j}$ is the localized

emission factor of pollutant $j$ for vehicle category $c$, in unit of grams per kilometer (g km$^{-1}$). $TF_{c,h,j}$ is the traffic flow of vehicle category $c$ on road link $l$ at hour $h$, in unit of vehicles per hour (veh h$^{-1}$). $L_l$ is the length of road link $l$, in unit of kilometers (km).

$$EF_{c,j} = BEF_{c,j} \times \varphi \times \gamma \times \lambda \times \theta, \tag{2}$$

Where, $BEF_{c,j}$ is the comprehensive baseline emission factor of pollutant $j$ for vehicle category $c$, in unit of grams per kilometer

(g km$^{-1}$). $\varphi$, $\gamma$, $\lambda$, and $\theta$ are the dimensionless environmental correction coefficient, traffic condition correction coefficient, deterioration correction coefficient, and vehicle usage conditions correction coefficient (*e.g.*, the load of diesel vehicles), respectively. All comprehensive baseline emission factors and correction coefficients were adopted from the Technical Guide for Compilation of Atmospheric Pollutants Emission Inventory from Road Motor Vehicles (Trial) (MEE, 2014).

To achieve accurate emission calculations, the traffic flows for fuel light- and heavy-duty vehicles were allocated to eight specific types of vehicles, including light-duty passenger vehicles (LDPVs), middle-duty passenger vehicles (MDPVs), heavy-duty passenger vehicles (HDPVs), light-duty trucks (LDTs), middle-duty trucks (MDTs), heavy-duty trucks (HDTs), public buses, and taxis. This classification followed the national standard (GA802-2019) (see Table S1). The distribution coefficients for the eight types of vehicles were referred to the hourly vehicle proportions from field surveys on different rods in urban Jinan in our previous study (see Table S2) (Wang et al., 2024). In addition, environmental correction was conducted mainly based on temperature and humidity which varied largely from season to season, as detailed in Tables S3–S6. As to the traffic condition correction, coefficients were determined based on the average vehicle speed intervals (see Tables S7–S8). Different types of road segments, as well as the same type of road segments during peak and off-peak hours, were assigned with differential vehicle speed intervals based on real-time road conditions from Gaode Maps (see Table S9). Note that new energy vehicles were deemed zero-emission in this study and the evaporative emissions of HC were excluded due to the complex sensitivities to fuel properties and environmental conditions (Jiang et al., 2021).

To present spatially continuous vehicle emission maps, the spatial interpolation of hourly average emission intensities was conducted to fill in the gaps between discrete monitoring points. Compared to directly filling the entire road segments with a single value, spatial interpolation allowed for the determination of pollutant emission rates at any point along the road, thereby generating emission maps that were more representative of real-world conditions. The nearest neighbor interpolation was uniformly selected for all four air pollutants due to its maximal preservation of the original emission data at the monitoring points. For a given point with unknown data, the nearest neighbor interpolation does not create a new value, but set by replicating the value of the known point located at the shortest distance (Olivier and Hanqiang, 2012). The proposed method demonstrated high performance in terms of achieving zero error at the monitoring points compared to the other interpolation methods. However, the continuity of the nearest neighbor interpolation results is limited, leading to noticeable step effect. Therefore, Gaussian smoothing was further applied to achieve smooth data by convolving a Gaussian kernel. The Gaussian kernel is a two-dimensional Gaussian function matrix, whose shape is determined by the standard deviation $\sigma$ (see Eq. 3) (Song et al., 2022).

$$G(x,y) = \frac{1}{2\pi\sigma^2} \exp(-\frac{\Delta x^2 + \Delta y^2}{2\sigma^2}) \tag{3}$$

Where, $G(x,y)$ is the weight of point $(x, y)$, $\Delta x$ and $\Delta y$ are the distances from the center point in the $x$ and $y$ directions, respectively, and $\sigma$ is the standard deviation and determines the degree of smoothing. Through the convolution operation, the value of each central point was updated using the weighted average of the surrounding data points with the weights provided by the Gaussian kernel. This process effectively reduced the noise in the emission map while preserving important features. Finally, our interpolation model not only improved the resolution of on-road vehicle emission mapping, but also smoothed the irregular variations caused by outliers, making the map more readable and interpretable. In addition, data pivoting was used to display aggregated values in the two-dimensional grids. By summarizing and analyzing the data under different situations such

as time periods, meteorological conditions, air pollutants, and future scenarios, the distribution patterns, variation trends, and relationships could be revealed.

## 2.4 Temporal and spatial clustering analyses on variation patterns

Spatiotemporal clustering analyses can reveal the variation patterns of vehicle emissions at different spatial and temporal scales, which is crucial for understanding the hotspot distribution and the dynamic change. In this study, time series clustering was used to identify the diurnal variation patterns of vehicle emissions over different types of roads (Tavakoli et al., 2020; Camastra et al., 2022; Barreto et al., 2023). The emissions of $NO_x$, CO, HC, and $PM_{2.5}$ were selected as the feature columns with 1 hour as the time step. With consideration of the long duration of the study (nearly a year) and the large number of monitoring points (1189), we simplified the data structure through averaging to avoid the curse of dimensionality. At first, the entire dataset was grouped by point and hour. Then, we calculated the hourly average values for each point and discarded incomplete time series. Finally, 1158 multidimensional time series were obtained, comprising of feature columns and a time column with a length of 24 hours. These multidimensional arrays were further normalized and utilized as inputs representing the original data, with the Euclidean distance employed as the metric to measure the distances between data points. A commonly used clustering algorithm, namely K-means (Mac Queen, 1967), was applied to group the time series into different clusters based on the distance metric results, with each cluster representing a set of data points exhibiting similar diurnal variation pattern. Specifically, the multi-dimensional data were clustered into $K$ clusters. Initially, $K$ centroids were selected randomly. Each data point was assigned to the nearest centroid based on the Euclidean distance. Then, the centroids were iteratively updated by calculating the mean of all data points assigned to each cluster (Boleti et al., 2020). The squared error ($\varepsilon$) between the centroid $\mu_k$ and the data point $x_i$ was calculated as shown in Eq. 4.

$$\varepsilon = \sum_{i=1}^{n} \sum_{k=1}^{K} \left\| x_i - \mu_k \right\|^2, \tag{4}$$

Where, $x_i$ is a data point, $\mu_k$ is the centroid of cluster $k$, and $\left\| x_i - \mu_k \right\|^2$ is the Euclidean distance between $x_i$ and $\mu_k$. By minimizing $\varepsilon$, the K-means algorithm can find the optimal centroid positions, such that the distance from each data point to its corresponding cluster centroid is minimized as much as possible. This process will repeat until the centroids no longer change significantly, indicating convergence. The Silhouette Coefficient (SC) was used to assess the performance of the clustering and determine the optimal clustering parameters (Rousseeuw, 1987). The number of clusters with the largest SC was considered as the most representative (Choi et al., 2024). In addition, under the determined number of clusters, the optimal choices for parameters such as the random seed number were identified through grid search.

Hot spot analysis calculates Getis-Ord Gi* statistics to identify statistically significant clusters of high values (hotspots) and low values (cold spots), thereby revealing spatial patterns of data aggregation. The $G_i^*$ statistic returned for each feature in the dataset is a $z$-score (see Eq. 5) (Ord and Getis, 1995).

$$G_i^* = \frac{\sum_{j=1}^{n} w_{i,j} x_j - \overline{X} \sum_{j=1}^{n} w_{i,j}}{S\sqrt{\frac{\left[n\sum_{j=1}^{n} w_{i,j}^2 - \left(\sum_{j=1}^{n} w_{i,j}\right)^2\right]}{n-1}}}, \ \overline{X} = \frac{\sum_{j=1}^{n} x_j}{n}, \ S = \sqrt{\frac{\sum_{j=1}^{n} x_j^2}{n} - (\overline{X})^2} \tag{5}$$

For statistically significant positive $z$-scores, a larger $z$-score indicates more intense clustering of hot spots. Conversely, for statistically significant negative z-scores, a smaller $z$-score indicates more intense clustering of cold spots. If the $z$-score is close to zero, it indicates no significant spatial clustering. The Optimized hot spot analysis (Esri, n.d.) was chosen to automatically aggregate incident data, identify an appropriate scale of analysis, and correct for both multiple testing and spatial dependence, finally reducing false positives and improving the accuracy of statistical significance.

**2.5 Scenario design for new energy vehicle replacement**

Based on high-resolution mapping of vehicle emissions, the benefits of replacing internal combustion engine vehicles (ICEVs) with new energy vehicles for emission reductions can be directly assessed. NEVs are mainly classified into battery electric vehicles (BEVs), plug-in hybrid electric vehicles (PHEVs), and fuel cell vehicles (FCVs) (Xie et al., 2024). Except for PHEVs, which emit pollutants in hybrid mode, all other NEVs produce no pollutants during driving. As to PHEVs, they only account for a relatively small proportion within NEVs (less than 20%) and they primarily operate in electric mode in short-distance driving. Since it cannot distinguish specific types of NEVs during traffic monitoring, in this study all NEVs are considered zero-emission for simplification and uniformity. The scenario design referenced the literature review in an existing study on the environmental benefits of NEVs (Peng et al., 2021) and some adjustments were made to fit the situation in the main urban area of Jinan. Here, four scenarios of NEV penetration were set up (see Table 1), mainly oriented to passenger vehicles (PVs), trucks, buses, and taxis. Note that there was only limited research on the future NEV penetration in MDTs and HDTs, mainly due to the challenges to meet the demand of relatively long driving ranges and to address the problems of high costs for large-capacity rechargeable batteries and charging infrastructure (Liang et al., 2019; Secinaro et al., 2022). The government is encouraging the promotion of new energy MDTs and HDTs (China State Council, 2024), with the possibility of achieving zero-emission freight fleets in the future. However, at present most of the new energy trucks in cities are LDTs or sanitation vehicles. With consideration of the above situation, we made a bold assumption here, predicting that MDTs and HDTs will achieve a 50% penetration in the EHP scenario. The LP, IP, HP and EHP scenarios described the EV penetration ranges for PVs and LDTs (10%–80%), MDTs and HDTs (2%–15%), buses and taxis (80%–100%), as well as FCV penetration for MDTs and HDTs (2%–35%). It was noteworthy that currently the NEV penetration in Jinan had already reached the LP level and was transitioning towards IP. Particularly, the NEV penetration for public transit (buses and taxis) had exceeded 80% with the active promotion of new energy policies. Nevertheless, in other cities with limited electricity supply and fewer charging devices, the NEV penetration could be lower than in the LP scenario.

**Table 1.** New energy vehicle penetration scenarios.

| Scenarios | EV-PVs & LDTs | EV-MDTs & HDTs | FCVs-MDTs & HDTs | EV-Buses & Taxis |
|---|---|---|---|---|
| Low penetration (LP) | 10% | 2% | 2% | 80% |

| Intermediate penetration (IP) | 50% | 5% | 10% | 90% |
| High penetration (HP) | 80% | 9% | 18% | 100% |
| Enhanced high penetration (EHP) | 80% | 15% | 35% | 100% |

## 3 Results and discussion

### 3.1 Distribution of traffic flows in the main urban area

Traffic flow is a key factor affecting the on-road vehicular emissions, so it is crucial to comprehensively understand the variations of traffic flow (Deng et al., 2020). By using nearest neighbor interpolation, we generated high-resolution mapping to visualize traffic flows in the main urban area of Jinan (see Fig. 3a). The traffic flows at a fine scale (50 m × 50 m) exhibited obvious spatiotemporal heterogeneities. Temporally, there were apparent diurnal variations (peak and off-peak hours) and weekly variations (weekdays and weekends) (see Fig. S1). Specifically, the daytime traffic volumes were much higher than those at nighttime, accounting for approximately 81.2% of the total. Traffic flows remained very low from 0:00 to 5:00 (local time) and then started to increase at 6:00. They exhibited a bimodal pattern with two peaks appearing in morning (7:00 to 9:00) and late afternoon (17:00 to 19:00), with a midday valley occurring at 12:00. Particularly, during peak hours on weekdays, the average traffic flow was 2188 veh $h^{-1}$, which was 62.4% higher than the 24 h average. By comparison, the diurnal variation curve on weekends was smoother, with the morning peak delaying to 9:00 to 10:00, and the midday valley was less noticeable. In addition, weekday traffic volumes were slightly higher than those on weekends, with average traffic flows of 1367 and 1308 veh $h^{-1}$, respectively. Although commuting vehicles decreased on weekends, vehicles for leisure trips and other reasons increased, resulting in temporal dispersion. The decrease in traffic volume was mainly concentrated during the morning and late afternoon peaks. On average, the traffic flow in peak hours on weekends reached only about 90.4% of that on weekdays.

Spatially, traffic high-value zones, defined as road segments or clusters with traffic flow more than triple the average level (>4000 veh $h^{-1}$), spread in the main urban area of Jinan. As shown by red segments in Fig. 3a, linear high-value zones can be observed along expressways and arterial roads. Urban expressways and arterial roads carried nearly 94% of the total traffic flows on the road network, serving as the major conduits for commuter traffic. Particularly, expressways had very high traffic flows (2940 veh $h^{-1}$ on average), due to possessing multiple lanes or the combination of elevated and ground-level lanes. Arterial roads had moderate traffic flows, with an average value of 1093 veh $h^{-1}$. In contrast, residential areas and local streets showed quite low traffic flows (<500 veh $h^{-1}$), highlighting the disparity in traffic distribution. Additionally, the traffic flows in the central business districts were substantially higher than those at the margins of urban areas, increasing the likelihood of traffic high-value zones. Furthermore, the traffic flows sharply increased at intersections due to the temporary stoppages caused by traffic lights. It was noteworthy that when temporal peaks coincided with spatial high-value zones, the traffic flows of the high-value zones were further intensified, which was called as the "overlap effect of spatiotemporal peaks". During peak hours of traffic, the spatial distribution of high-value zones remained generally unchanged, but their extent expanded and traffic flows increased (see Fig. S2).

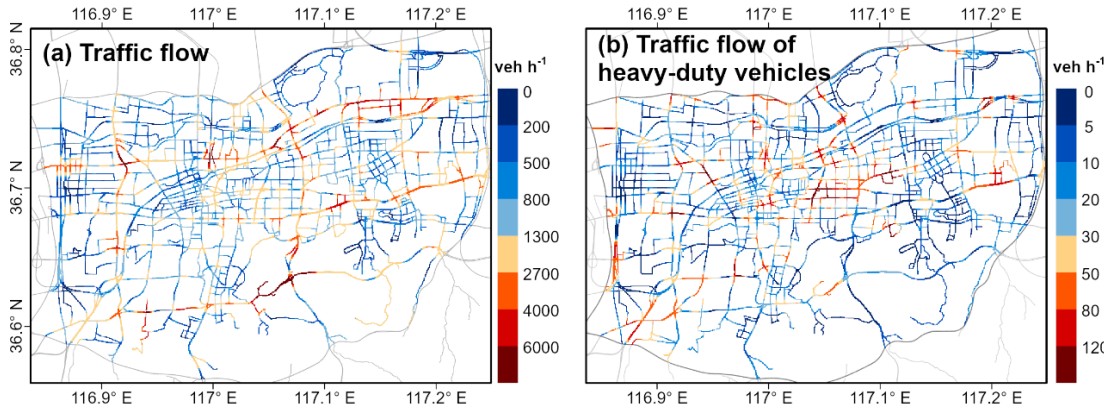

Figure 3. High-resolution mapping of hourly average traffic flows for (a) all on-road vehicles and (b) HDVs in the main urban area of Jinan.

In terms of vehicle composition, as shown in Fig. S3, due to the absolute dominance of LDPVs (approximately 90%), they largely determined the spatial and temporal distribution of traffic flows. It was worth mentioning that the proportion of NEVs in the on-road vehicles approached 18%. There was no significant difference in the fractions of various types of vehicles between weekdays and weekends, indicating relatively stable vehicle composition over the week. Except for MDTs and HDTs, all other types of vehicles primarily operated during daytime, showing a bimodal diurnal variation pattern. As to MDTs and HDTs, due to the stringent transportation management policy in Jinan (*i.e.*, banning MDTs and HDTs to enter the main urban area during peak hours and to enter the area within the Second Ring Road during off-peak hours), their temporal variations and spatial distributions were different from other types of vehicles. There were rare MDTs and HDTs other than certain municipal vehicles on road within the urban area during peak hours. Many MDTs and HDTs drove into the roads outside the Second Ring Road during off-peak hours, with increased traffic flows at night. In addition, the high-value zone distribution of heavy-duty vehicle flows was significantly different from that of total traffic flows (as shown in Fig. 3b). As mentioned above, the total traffic flows, dominated by LDPVs, reached their maxima on expressways. In contrast, HDVs mainly operated on arterial roads, with high-value zones appearing on densely populated arterial roads in the city center.

### 3.2 Variation characteristics of on-road vehicle emissions

The emissions of different air pollutants from on-road vehicles in the main urban area of Jinan were calculated. The hourly average emissions of $NO_x$, CO, HC, and $PM_{2.5}$ were 345.2, 789.7, 69.5, and 5.4 kg, respectively. There were large variation in the contributions of different types of vehicles to each air pollutant (see Fig. S4). Specifically, CO and HC were primarily contributed by LDPVs, accounting for over 60%, which was consistent with previous studies in China (Liu et al., 2018; Sun et al., 2021; Yang et al., 2019). In addition, LDTs also contributed large portions to CO (20%) and HC (15%). Since both LDPVs and LDTs mainly used gasoline fuel, it can be inferred that CO and HC were mainly emitted from gasoline vehicles. In contrast, HDVs (*e.g.*, HDTs, HDPVs, and buses) mainly used diesel fuel, and their shares in $NO_x$ and $PM_{2.5}$ emissions were much greater than those in CO and HC emissions. For $NO_x$, nearly all types of vehicles (except taxis) contributed significantly

to its emissions. Particularly, buses and HDTs were the largest contributors (approxiamte 60%) to $NO_x$ emission, even though they only made up less than 2% of the traffic volume. Surprisingly, LDPVs have become the largest contributor (38%) to $PM_{2.5}$ emissions, although their contribution was less significant in the past. On the one hand, as emission standards have become more restrictive, the differences in emission factors among LDPVs, HDPVs, HDTs, and buses have diminished (Huang et al., 2017; Sun et al., 2021). On the other hand, the volume of LDPVs is much higher than that of other vehicle types. Note that the contributions to air pollutant emissions from different types of vehicles varied among different cities in China (*e.g.*, Beijing, Nanjing, Chongqing, Foshan, etc.) (Wu et al., 2022; Zhang et al., 2018; Ding et al., 2021; Liu et al., 2018), mainly due to the differences in vehicle composition. For example, buses are the primary travel mode for citizens in the main urban area of Jinan. They have long routes, necessity to maintain low to moderate speeds with frequent stop-and-go movements, and relatively high emission factors, resulting in high contributions to the emissions of $NO_x$ and $PM_{2.5}$. Similarly, in the urban area of Beijing, buses contributed 30% of $NO_x$ emissions (Yang et al., 2019). Additionally, due to strict truck traffic restrictions in the urban area, the contribution of HDTs in this study was smaller than other studies involving intercity highways (Yang et al., 2019; Zhu et al., 2023).

Vehicle emissions in urban Jinan exhibited large temporal variations, mainly caused by changes in traffic flow and vehicle composition. There were similar diurnal patterns in the emissions of CO, HC, and $PM_{2.5}$, presenting two distinct peaks in peak hours, while $NO_x$ showed a broad peak during daytime with a small decrease in the midnoon (see Fig. 4). On the one hand, the vehicle emissions of CO, HC, and $PM_{2.5}$ were dominated by LDPVs. During the peak hours in the morning and late afternoon with the highest traffic flows, the hourly emission intensities of CO, HC, and $PM_{2.5}$ also reached their peaks, with averages of 1649.4, 165.0, and 10.8 g km$^{-1}$, respectively. These values were 2.5–2.9 times larger than their 24-hour average levels. On the other hand, $NO_x$ emissions were dominated by HDVs for passenger and freight transport. Among them, buses and HDPVs operated in close alignment with regular schedules and daily life, primarily concentrated during daytime. In contrast, HDTs commonly flooded into the urban area during off-peak hours and midnight period due to urban traffic restrictions, causing elevation of the $NO_x$ emissions from HDT during off-peak periods. As a result, $NO_x$ emissions were distributed throughout daytime, with less pronounced peaks during peak hours (see Fig. 4a). The hourly average emission intensities of $NO_x$ during peak hours was only 66.4% higher than the 24 h average. Generally, the unique traffic behaviors of HDVs led to disparate temporal patterns for air pollutant emissions.

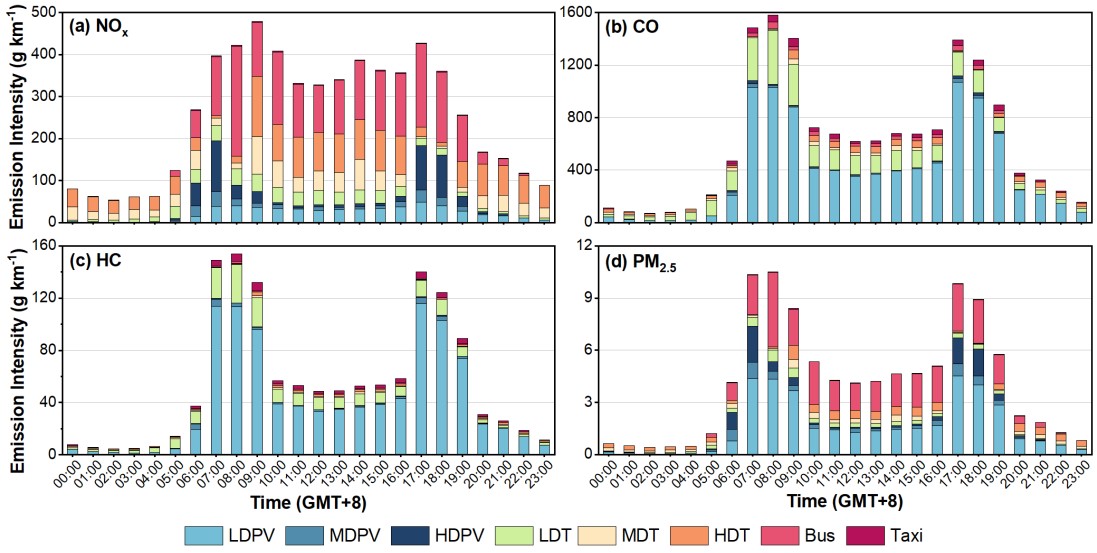

**Figure 4.** Average hourly emission intensities from different types of vehicles at traffic monitoring points for (a) $NO_x$, (b) CO, (c) HC, and (d) $PM_{2.5}$.

In addition, nearly one year of emission data enabled us to investigate the seasonal variations of vehicle emissions on urban roads. The monthly variation trends were generally similar for CO, HC, and $PM_{2.5}$ but a little different for $NO_x$, primarily determined by traffic flows and the meteorological conditions. As shown in Fig. 5, traffic activities were intensified in summer (especially in July) when compared with other seasons, leading to overall higher $NO_x$ emissions in summer. On the one hand, people preferred using cars rather than walking or cycling in the hot summer. On the other hand, summer was the peak tourist season and many out-of-town vehicles contributed to the increased traffic volume in Jinan. Notable reductions in traffic flows and pollutant emissions were observed during Chinese official holidays, especially during the National Dayand Spring Festival holidays (Sep. 30[th] to Oct. 6[th] and Feb. 9[th] to 17[th]). It was because human travels and commercial activities decreased, causing a sharp drop in gasoline vehicles (mainly private cars) and diesel vehicles (mainly trucks) in particular at the beginning of the holiday, followed by a gradual return to normal levels. Futhermore, the seasonal differences in vehicle emissions among different pollutants were quite pronounced due to meteorological conditions. $NO_x$ emissions were higher in summer than other seasons, partly owing to the high temperature in hot season. In contrast, HC and $PM_{2.5}$ emissions peaked in winter, to a large degree associated with the low temperature in cold season. High temperatures in summer affected engine combustion efficiency, leading to increased $NO_x$ emissions. HC and $PM_{2.5}$ emissions, however, were more sensitive to low-temperature conditionss and increased during cold starts due to incomplete fuel combustion. Note that the diurnal variation patterns of vehicle emissions across different seasons were similar, with high emissions occurring during daytime. Emissions in the morning (6:00-11:00) and afternoon (12:00-17:00) accounted for 41.0% and 33.2% on average, respectively, while those in the early morning (0:00-5:00) and evening (18:00-23:00) only accounted for 4.9% and 20.9%, respectively. Overall, the above seasonal and diurnal variation characteristics of air pollutant emissions from on-road vehicles provide scientific basis for accurate air quality modeling and refined urban air pollution control.

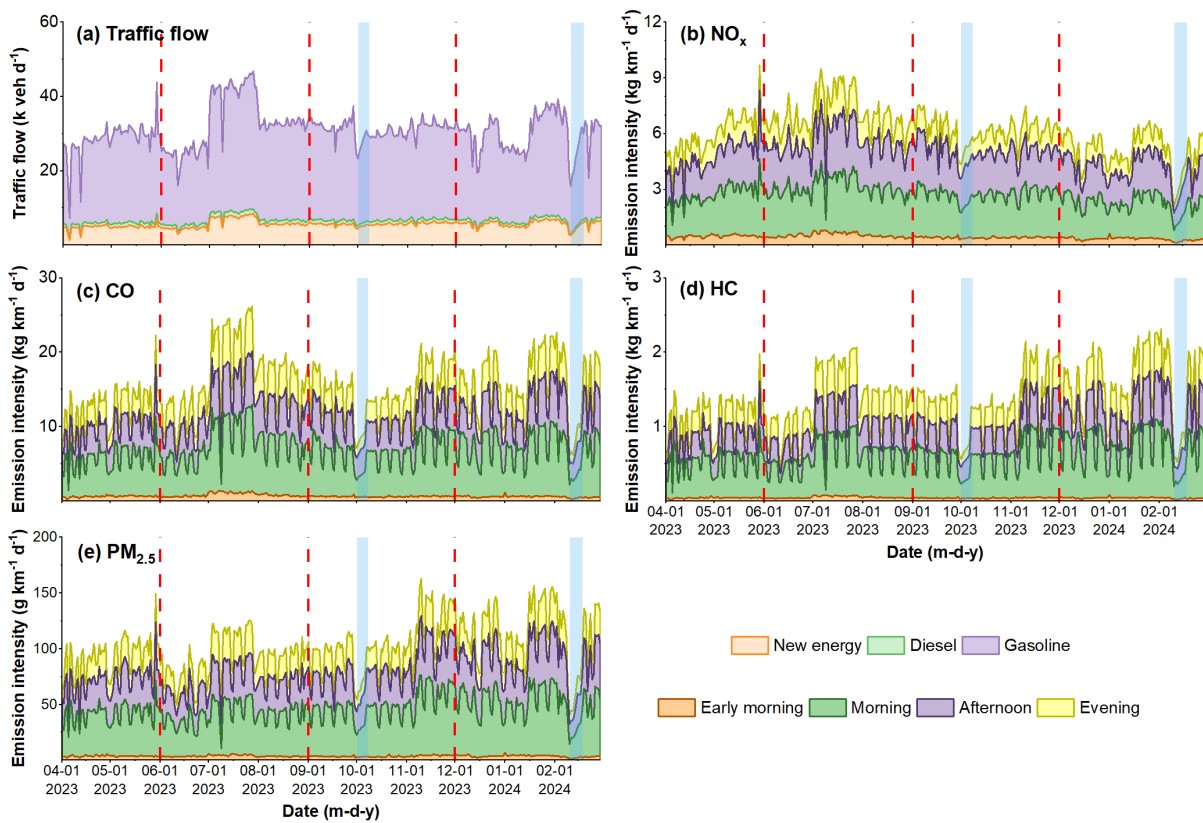

**Figure 5.** Daily average of (a) traffic flows for different types of vehicles and emission intensities of (b) NO$_x$, (c) CO, (d) HC, and (e) PM$_{2.5}$ during different time periods of the day. Red dashed lines represent the division by seasons, while blue bars represent official holidays.

Spatially, the distribution characteristics of air pollutant emissions from on-road vehicles were strongly associated with traffic flow distributions (see Fig. 6). This variation pattern to some degree differed from previous results, which generally show a decrease from the urban center to the periphery in a radiating structure (Zhang et al., 2018; Yang et al., 2019). Firstly, HDVs dominated NO$_x$ and PM$_{2.5}$ emissions, and their spatial distributions exhibited similar characteristics to the distribution of heavy-duty vehicle flows, with high-emission zones (hourly emission intensity > 850 g km$^{-1}$) primarily appearing on arterial

roads in the city center. LDVs dominated CO and HC emissions, thus considerably influencing their spatial distributions, with linear high-emission zones observable along urban expressways. Therefore, influenced by the dominant types of vehicles, the spatial distributions of vehicle emissions in urban Jinan varied for different air pollutants, which is consistent with the previous study by Sun et al. (2021). Secondly, grids with high emissions were predominantly located at urban expressways or the intersections of arterial roads. In contrast, grids with low emissions were generally situated on residential streets and the urban

edges with relatively low traffic volumes. The average traffic flows on the urban expressways, highways, arterial roads, secondary roads, and local streets followed a descending order, with the emission intensities following the same decreasing trend. For instance, the calculated hourly average NO$_x$ emission intensities on the urban expressways, highways, arterial roads, secondary roads, and local streets were 485.1, 303.4, 240.8, 84.2, and 74.6 g km$^{-1}$, respectively. Among the four types of roads,

arterial roads were of the longest length with the largest traffic volume, accounting for 48.4% of the total, while the corresponding emissions contributed 54.9% of the total emissions. Urban expressways carried 45.6% of the total traffic volume, but their emissions amounted to only 38.7%. The primary reason for this discrepancy was the difference in vehicle compositions across different types of roads, *i.e.*, the volume of HDVs on arterial roads was up to 35.6% and higher than that on urban expressways. As a result, although vehicle compositions were independent of the distribution of the traffic flows, they largely affected the emission distribution characteristics, particularly over fine-scale areas. In addition, Figure 6 shows that high emissions frequently appeared in intersections, with emission intensities radiating from the intersection to the surrounding roads. This feature was rarely presented in other high-resolution vehicle emission mappings (Jiang et al., 2021; Wu et al., 2022), but in this study, interpolation made the differences between intersections and road segments more pronounced (see Fig. S5). The emissions within 1 km of an intersection varied significantly, by a factor of 1.4–3.

Figure 7 takes $NO_x$ as an example to show the spatial distributions of vehicle emissions during different time periods in the main urban area of Jinan, aiming to explore the emission patterns under the joint influence of temporal and spatial characteristics. Firstly, the vehicle emissions during daytime were significantly higher than those at nighttime, with daytime emissions contributing more than 74% of the total $NO_x$ emissions. During the early morning, there were virtually no high value distributed across urban Jinan (as shown in Fig. 7c). Secondly, the emissions during the short peak hours accounted for approximately 37% of the daytime emissions on weekdays, with high-emission zones accounting for 14.7%. Note that the "overlap effect of spatiotemporal peaks" observed in traffic flow mapping could also be seen in the mapping of vehicle emissions. For example, the spatial distribution of vehicle emissions during peak hours (see Fig. 7a) remained generally unchanged when compared to off-peak hours (see Fig. 7b). However, the high-emission zones (red lines in Fig. 7a and b) expanded significantly on the original basis, with the hourly average emission intensity increasing by 1150 g km$^{-1}$. In contrast, roads with initially low emissions (blue lines in Fig. 7a and b, hourly emission intensity < 100 g km$^{-1}$) only showed an increase of 63.4 g km$^{-1}$ in hourly emission intensity during the peak hours, with most still remaining at low levels. Thirdly, the spatial distributions of vehicle emissions on weekdays and weekends were generally consistent, with slightly higher emission levels on weekdays than weekends (292.3 vs. 247.9 g km$^{-1}$ in the hourly average emission intensity) (see Fig. 7f). All in all, the above spatial variation patterns of $NO_x$ emissions from on-road vehicles were closely related to regular commuting and the lifestyle of residents, and the same applied to the other three pollutants (see Figs. S6-S8).

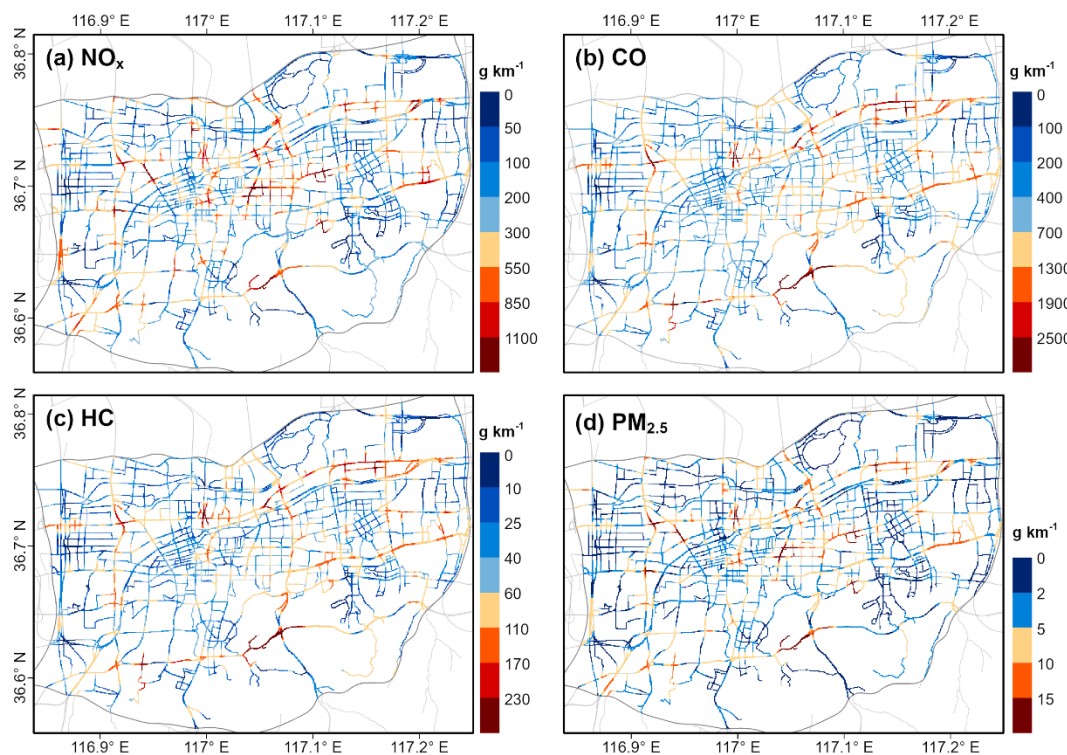

**Figure 6.** High-resolution mapping of hourly average vehicle emission intensities of major pollutants including (a) NO$_x$, (b) CO, (c) HC, and (d) PM$_{2.5}$ in the main urban area of Jinan.

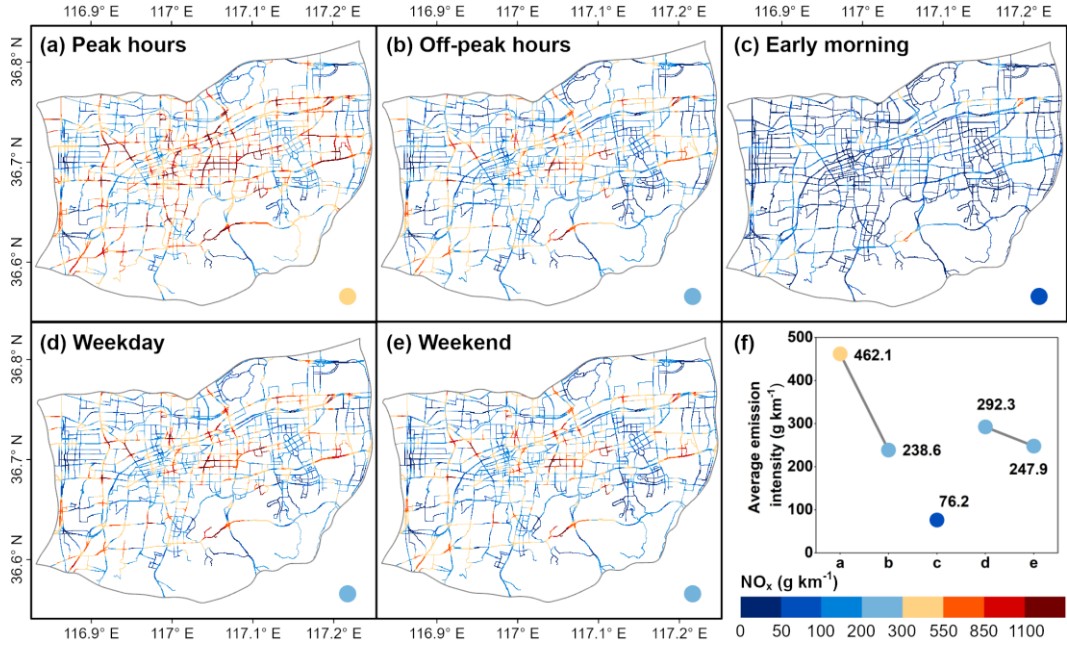

**Figure 7.** High-resolution mapping of on-road vehicle NO$_x$ emissions during (a) peak hours, (b) off-peak hours, (c) early morning, (d) 405 weekday, and (e) weekend and (f) average emission intensities of NO$_x$ during each time period.

### 3.3 Temporal and spatial cluster characteristics of vehicle emissions

To further understand the spatiotemporal variation behaviors and the main influencing factors of vehicle emissions, temporal and spatial clustering analyses were conducted with the hourly vehicle emission data for four pollutants. Based on the results from time series clustering, the optimal number of clusters ($K$) was determined as two with the highest silhouette coefficient and two distinct diurnal variation patterns. As shown in Fig. 8a and c, CO and HC emissions in Cluster 1 were concentrated during traffic peak hours in the morning and late afternoon, with emission intensities dropping sharply after the peaks. Additionally, $NO_x$ and $PM_{2.5}$ emissions in Cluster 1 also showed two distinct peaks during peak hours. In contrast, the diurnal emission plots for four air pollutants in Cluster 2 were relatively smooth, with no distinct peaks, and the differences were only evident between daytime and nighttime (see Fig. 8b and d). The fluctuation degrees of diurnal emissions for each pollutant in the two clusters were calculated by using the coefficient of variation (CV) and it was found that the average CV of Cluster 1 was 82.5%, much higher than that of Cluster 2 (55.5% on average). Furthermore, the emission levels of the two clusters were similar, with $NO_x$ and CO emissions in Cluster 2 slightly higher than those in Cluster 1 (by 26.5% and 12.7%, respectively). Figure 8e visually shows the spatial distributions of the two clusters, showing the areas affected by different emissions patterns. It was evident that most traffic monitoring points on urban expressways and highways belonged to Cluster 2. These roads were mainly used for long-distance travelling and fast passages, and did not experience large-scale traffic congestion during peak hours. In addition, vehicles on elevated roads maintained high speeds and uniform traffic flows, with few stops due to traffic lights or congestion, resulting in evenly distributed emissions throughout the day. Some residential roads with low traffic volumes were also classified into Cluster 2. Instead, Cluster 1 mainly distributed on arterial roads, intersections, and near commercial areas within the city, where vehicles were dense during peak hours, leading to sharp increases in emissions due to the "overlap effect of spatiotemporal peaks". As a result, the diurnal variation of vehicle emissions was not uniform across roads with different characteristics, indicating spatial instability. This has not received much attention in previous studies, which typically considered the entire region as a whole and revealed only bimodal patterns (Jiang et al., 2021; Ding et al., 2023).

The findings on the diurnal variation patterns of vehicle emissions and the corresponding spatial distribution characteristics provide scientific recommendations for emission reduction measures and resident travel. For example, controlling the traffic volume during peak hours is required on arterial roads to reduce traffic congestion and high emissions of air pollutants. On expressways, ensuring unblocked traffic and improving transit efficiency are particularly important. Regarding resident travel, it is recommended to avoid major arterial roads and other easily congested areas during peak hours by traveling earlier or later. Residents can also select public transportation, bicycles, or walking as alternative modes of travel. In addition, choosing expressways with high traffic efficiency is advisable. While driving on expressways, maintaining a steady speed, and avoiding frequent lane changes and sudden acceleration or braking can improve fuel efficiency and reduce pollutant emissions.

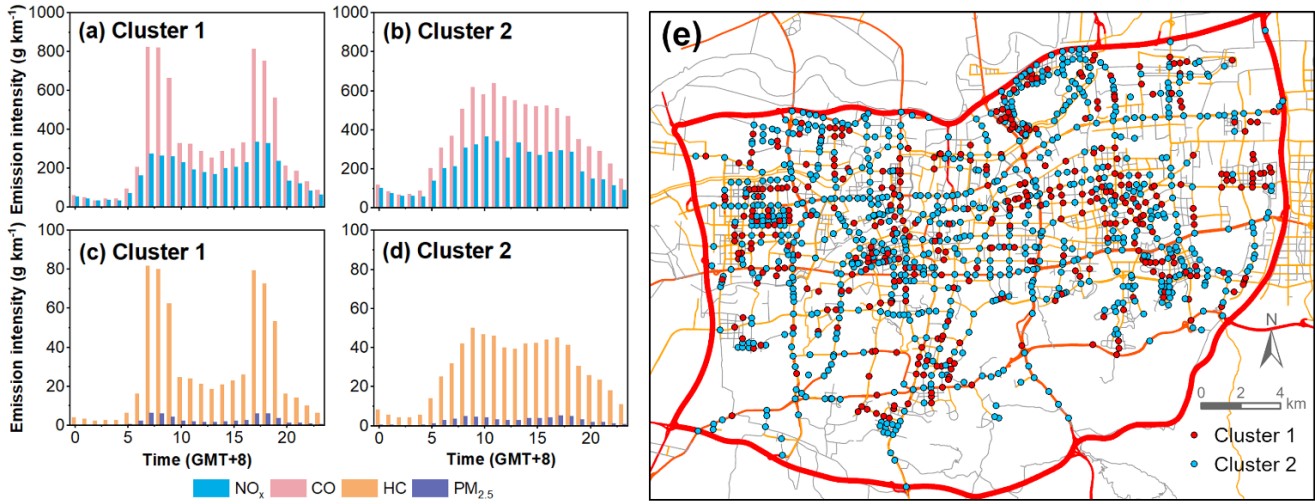

**Figure 8.** Hourly average emission intensities of NO$_x$ and CO for (a) Cluster 1 and (b) Cluster 2 and HC and PM$_{2.5}$ for (c) Cluster 1 and (d) Cluster 2 and (e) the distributions of traffic monitoring points belonging to different clusters in the main urban area of Jinan.

Spatial clustering characteristics of vehicle emissions were obtained through hotspot analysis by calculating the Getis-Ord Gi* statistics. The locations and potential causes of clustering for high or low-value features were identified at a fine-scale (see Fig. 9). The emission hotspots for different pollutants were likely driven by different factors, including high traffic volumes, concentrated HDVs, and road intersections (see Fig. S9). Specifically, NO$_x$ and PM$_{2.5}$ emission hotspots were dominated by HDVs, clustered along densely trafficked arterial roads in the city center, while the cold spots mostly appeared in residential areas (see Fig. 9a and d). There was huge difference in the emissions between hotspots and cold spots. For example, the hourly average NO$_x$ emission intensity on Heping Road (hotspot area) was 2557.9 g km$^{-1}$, which was 172 times higher than that on Xingfusi Road (cold spot area, the 14.9 g km$^{-1}$ on average). In addition, CO and HC emissions usually peaked on urban expressways with high traffic volumes, forming linear hotspot areas (see Fig. 9b and c). Their highest emission was found on the South Second Ring Elevated Road, which had the highest traffic volume (see Fig. 3a). Furthermore, the emission hotspots remained spatially stable, showing similar from day to day. However, due to the "overlap effect of spatiotemporal peaks", the emission intensities at hotspot areas varied over time periods during the day, with peaks typically appearing around 08:00 in the morning and 18:00 in the late afternoon on weekdays.

Overall, both the temporal and spatial clustering analyses suggest that only a small number of vehicles and roads contributed very high emissions, while the majority exhibited relatively low emissions. The phenomenon that high-emission vehicles and roads made notable contributions to the total emissions is consistent with the finding reported by Böhm et al. (2022). Therefore, effective policies for vehicle emission reduction should primarily focus on key time periods, key areas, and key types of vehicles. For instance, differentiated traffic restriction measures can be implemented based on the dual-peak emission characteristics in the morning and late afternoon. Introducing new energy public transportation in identified high-emission time periods and areas will reduce the use of private vehicles. Utilizing real-time traffic data to intelligently adjust signal timing can adapt to peak and off-peak traffic periods, thereby minimizing congestion and pollutant emissions (Yang et

al., 2020). Implementing low-emission zone policy to restrict high-emission vehicles from entering the city center is also recommended.

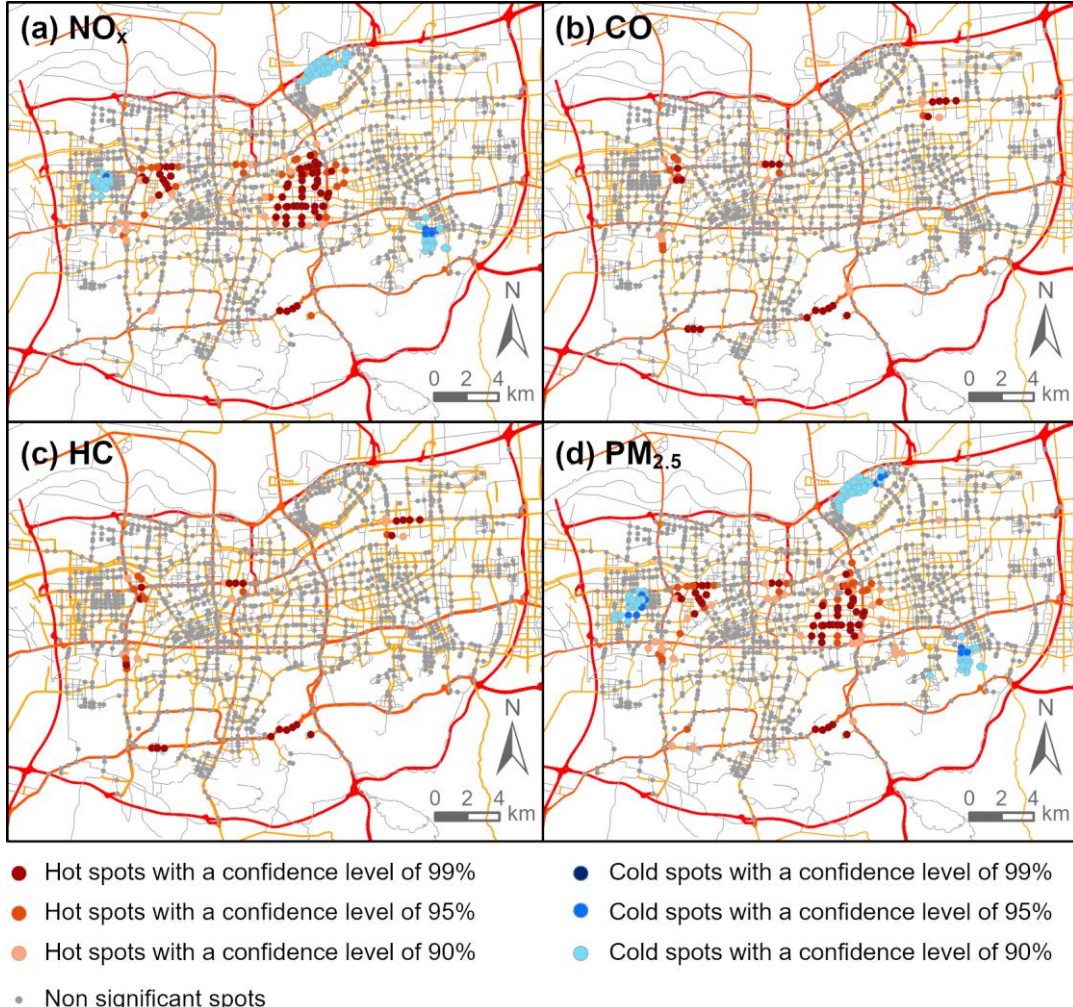

**Figure 9.** Spatial distributions of hot and cold spots of on-road vehicle emissions for (a) $NO_x$, (b) CO, (c) HC, and (d) $PM_{2.5}$.

### 3.4 Impacts of new energy vehicle replacement

Due to policy support, environmental benefits, technological advancements, and cost reductions, the population of new energy vehicles in China has been growing rapidly in recent years (Zhang et al., 2023). According to statistics (MPS, 2024), 7.43 million NEVs were newly registered in 2023, accounting for 30.2% of the total number of newly registered vehicles. NEVs can reduce the emissions of air pollutants from vehicles and subsequently improve urban air quality. Therefore, it is necessary to assess the specific emission reduction benefits brought by vehicle electrification with assumptions of suitable future penetrations of NEVs. Here in, the current emission levels in the main urban area of Jinan were used as the base case. Since the LP scenario was very close to the current situation of the city, it was not presented in details in this study. Three scenarios

including IP, HP, and EHP reflect discrepant electrification penetration for different types of vehicles. Specifically, the IP and HP scenarios comprehensively increase the penetration of NEVs across various vehicles. Particularly, large fractions of LDVs will be replaced with NEVs as they contributed significantly to on-road vehicle emissions currently. For the HDVs, which have high emission factors, the electrification of buses has been successful, whereas the transformation of trucks has faced challenges and proceeded slowly. In addition, we designed the EHP scenario, which further increase the NEV penetration of MDTs and HDTs to achieve more appreciable emission reduction benefits (Böhm et al., 2022; Tian et al., 2022).

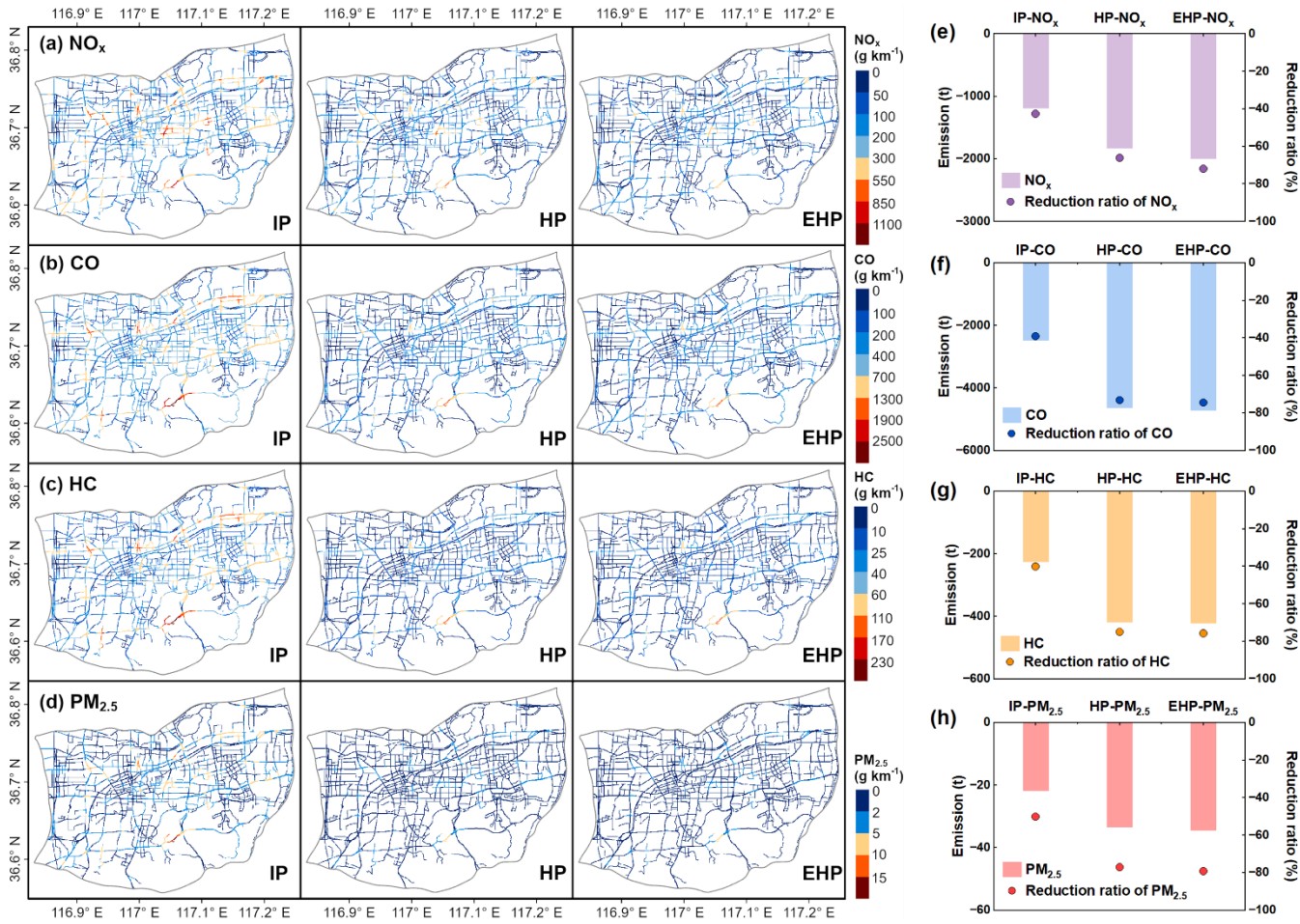

**Figure 10.** On-road vehicle emission mappings for (a) NO$_x$, (b) CO, (c) HC, and (d) PM$_{2.5}$ simulated under different new energy penetration scenarios and emission reductions for (e) NO$_x$, (f) CO, (g) HC, and (h) PM$_{2.5}$ under each scenario relative to the base case during the study period. Bars indicate the emission reduction amounts and dots indicate the percentage reduction.

Figure 10 shows the on-road emission intensities for NO$_x$, CO, HC, and PM$_{2.5}$ in the main urban area of Jinan under different future scenarios and the reductions compared with current emissions. Specifically, under the IP scenario, vehicle emissions decreased by large degrees, *i.e.*, 42.7% for NO$_x$, 39.1% for CO, 40.3% for HC, and 50.1% for PM$_{2.5}$ when compared with the present emission intensities, and most of the emission hotspots disappeared. As to the HP scenario, which involved a

further increase in NEV penetration, this led to significant additional reductions in emissions particularly on the urban expressways and arterial roads, *i.e.*, $NO_x$ decreased by 23.4%, CO by 34.0%, HC by 34.8%, and $PM_{2.5}$ by 27.0% relative to the IP scenario. At that time, emission hotspots on all roads were virtually eliminated, indicating the positive impact of comprehensive vehicle electrification on road vehicle emissions, especially in the city center. Surprisingly, in the EHP scenario, where we increased the NEV penetration of MDTs and HDTs to 50%, emissions were only further reduced by an average of about 2.5%. Among them, $NO_x$ emissions, which were most affected by MDTs and HDTs, showed the highest reduction of 5.9%. As mentioned above, within the traffic restrictions in the main urban area, the proportion of MDTs and HDTs in urban areas was very small (only 2%) and their emissions only contributed a small fraction to the total vehicle emissions. Consequently, the reduction in on-road vehicle emissions was not significant under the EHP scenario.

Note that when evaluating the impact of NEV penetration on on-road vehicle emissions, our calculations and analyses were based merely on exhaust emissions, focusing on the differences in exhaust emissions between ICEVs and NEVs. However, a substantial portion of PM emissions are contributed by non-exhaust sources (e.g., road dust, brake wear, and tire wear) (Zhang et al., 2020). Additionally, NEVs are typically heavier than conventional vehicles, which may lead to higher non-exhaust emissions (Timmers and Achten, 2016; Liu et al., 2021). Therefore, if the contribution of non-exhaust emissions was also taken into account, the estimated benefits of increased NEV penetration on reducing $PM_{2.5}$ emissions would be lower than suggested by our current analysis.

## 3.5 Comparison with other vehicle emission inventories

To verify the reliability of the high-resolution emission inventories of on-road vehicles in the main urban area of Jinan obtained in this study, comparison with other limited emission inventory in previous studies were conducted. Firstly, Feng et al. (2023) established a high-resolution vehicle emission inventory for $NO_2$ and CO in Jinan for the year of 2021 by using a top-down approach with a resolution of 1 km × 1 km. Their results showed similar spatial distribution patterns to our study, *i.e.*, the high-emission zones were concentrated in the city center or near high-grade roads, with notably higher pollutant emissions in Lixia District than other areas. Nevertheless, due to the lack of dynamic traffic data, their inventory failed to present the fine-scale gradients of on-road vehicle emissions. Secondly, from the Multi-resolution Emission Inventory for China (MEICv1.4) with a low resolution of 0.25° × 0.25° for 2020 Zheng et al. (2014), there were only two grids in the main urban area of Jinan (see Fig. S10). When compared with the re-aggregated total emissions from gasoline and diesel vehicles in the MEICv1.4, the monthly average air pollutant emissions of on-road vehicles obtained in our study were significantly lower, with ratios of 58.1% for $NO_x$, 28.5% for CO, 12.3% for HC, and 51.2% for $PM_{2.5}$. The main reason for the apparently low HC was that the evaporative emissions of HC were excluded in this study. In addition, the continuous implementation of emission reduction measures including elimination of high-emission vehicles, improvement in fuel quality and emission standards, and promotion of NEVs in the past three years also contributed to the lower emissions in our study.

## 4 Summary and conclusions

520 This study developed a high-resolution on-road vehicle emission inventory for the main urban area of Jinan by using a bottom-up approach based on a large amount of real-world traffic data collected in real-time from more than 1000 traffic monitoring points. Fine-scale traffic flows and vehicle compositions almost over one year were gathered with extensive traffic cameras and field surveys to calculate the emission intensities from different types of vehicles. Multiple big data methods were utilized to demonstrate the temporal and spatial variation patterns of vehicle emissions with the massive traffic dataset. Specifically,

525 nearest neighbor interpolation and Gaussian smoothing were used to fill in spatial data gaps to obtain spatially continuous vehicle emission maps in ultra-high resolution of 50 m. Through time series clustering and hotspot analysis, the spatiotemporal clustering characteristics of vehicle emissions were analyzed and travel recommendations for residents were provided. The benefits of vehicle emission reduction with increased NEVs adoption in the future were predicted at three potential scenarios with different NEV penetration.

530  Results show that the daily average on-road vehicle emissions in the main urban area of Jinan were 8.28, 18.95, 1.67, and 0.13 t for $NO_x$, CO, HC, and $PM_{2.5}$, respectively. Among different types of vehicles, the contributions of HDTs to pollutant emissions were relatively small (2%–23%), due to strict traffic restrictions in the main urban area. The contribution of buses was a little large (1%–34%), demonstrating the importance of further promoting the electrification of public transportation in Jinan. The variation of vehicle emissions was strongly affected by traffic activities, primarily occurring during daytime, with

535 the highest emission intensities during peak hours (2.5–2.9 times as high as the hourly average levels). The emissions of air pollutants of CO, HC, and $PM_{2.5}$ exhibited distinct bimodal diurnal patterns, primarily contributed by LDPVs with contributions of 38%–74%. In contrast, 66% of $NO_x$ emissions were caused by HDVs, which were distributed throughout the day with less pronounced peaks during peak hours. The emission hotspots of CO and HC were linearly distributed along urban expressways with high traffic volumes, whereas those of $NO_x$ and $PM_{2.5}$ were mainly concentrated on arterial roads in the city

540 center where there were more HDVs. Particularly, "overlap effect of spatiotemporal peaks" was observed in on-road vehicle emissions in urban Jinan. When the temporal peaks coincided with the spatial hotspots, the emissions were further intensified. During the peak hours, the high-emission zones expanded significantly, with the hourly average $NO_x$ emission intensity increasing by 1150 g km$^{-1}$, while the low-emission zones only showed an increase of 63.4 g km$^{-1}$. In addition, on-road vehicle emissions exhibited notable seasonal differences, with higher $NO_x$ emissions in summer but higher HC and $PM_{2.5}$ emissions

545 in winter. Furthermore, the simulations of NEV penetration scenarios indicate that the electrification of the vehicles has large impacts on vehicle emissions and their spatial patterns. There were 40%–80% reductions in emissions and most hotspots disappeared in the future. These results not only demonstrate the potential of fleet electrification in emission reduction, but also provide a scientific basis for formulating more precise emission reduction strategies.

  More importantly, the framework of high-resolution vehicle emissions developed in this study applies to almost any other

550 city. Once traffic monitoring and road network data with full coverage are provided, it can support decision-makers in implementing emission reductions, improving citizen welfare, and designing strategies for more sustainable cities. As more

traffic data becomes available in the future, research work can be extended to include the surrounding suburban and rural areas. Additionally, the potential of big data techniques in establishing emission inventories can be further explored, to establish a dynamic big data platform that integrates multiple sources such as traffic cameras, low-cost sensors, GPS data, and open-

555 source congestion maps. Machine learning (ML) can further enhance the framework by dynamically optimizing emission factors based on traffic patterns, vehicle types, and meteorological conditions, thereby achieving real-time traffic data processing and dynamic updates of vehicle emission inventory.

**Data availability.** The meteorological data, including temperature and humidity, used in this study were obtained from the fifth-generation European Centre for Medium-Range Weather Forecasts (ECMWF) reanalysis data (ERA5;

https://doi.org/10.24381/cds.adbb2d47, Hersbach et al., 2023). All other data presented and used throughout this study can be accessed from the following data repository: https://doi.org/10.17632/24t54p6rj2.1 (Wang, 2024).

**Supplement.** The supplement includes ten figures (Figure S1-S10) and three tables (Tables S1-S3) related to the paper.

**Author contributions.** XW designed the research, supported funding, and edited the manuscript. HW, BZ and PL prepared the traffic dataset. YW processed and analyzed data, plotted the figures, and drafted the manuscript. SS, LX, QZ and QW

contributed to the scientific discussions. All authors contributed to the discussions of the results and the refinement of the manuscript.

**Competing interests.** The authors declare that they have no conflict of interest.

**Acknowledgements.** This study was supported by the National Natural Science Foundation of China (Nos. 42361144721, 42377094). The authors would like to express gratitude to ERA5 for providing meteorological data. The contents of this paper

are solely the responsibility of the authors and do not necessarily represent official views of the sponsors or companies.

**Financial support.** This study was supported by the National Natural Science Foundation of China (Nos. 42361144721, 42377094).

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
