# Peer review of "High-resolution mapping of on-road vehicle emissions with real-time traffic datasets based on big data"

_EGUsphere, 2024_

## Author Comment (AC1)

**Response to Commentsfrom Community #1 – Dr. Manmeet Singh**

**Manuscript:** egusphere- 2024-2791

**Title:** High-resolution mapping of on-road vehicle emissions with real-time traffic datasets based on big data

**Authors:** Yujia Wang et al.

**Corresponding author:** Xinfeng Wang (email: xinfengwang@sdu.edu.cn)

**General Comments:** The article "High-resolution mapping of on-road vehicle emissions with real-time traffic datasets based on big data" represents a significant advancement in urban emission inventorying by leveraging real-time data from traffic monitoring networks. This study, conducted in Jinan, China, presents a methodologically robust framework for capturing the spatiotemporal complexity of on-road emissions with unprecedented resolution. By integrating traffic camera networks with advanced big data methods, the authors developed a 50 m by 50 m, high-resolution map of emissions across pollutants, including $NO_x$, CO, HC, and $PM_{2.5}$.

The study is particularly innovative in its bottom-up approach, employing spatial interpolation methods to address the challenges p=osed by gaps between monitoring points. The use of Gaussian smoothing and nearest-neighbor interpolation effectively compensates for the spatial discontinuities typical in urban traffic networks. The authors also apply clustering techniques to analyze and interpret diurnal emission patterns, uncovering "hotspot" areas with temporal overlap during peak traffic periods. This attention to dynamic, real-world traffic flows yields an emissions inventory that is not only spatially continuous but also highly reflective of the real-world variability in urban emissions.

One of the key strengths of this paper lies in its interdisciplinary approach, combining atmospheric science, AI, and traffic monitoring technologies to generate actionable insights for urban policymakers. The authors demonstrate how time series clustering and hotspot analysis can aid in targeted interventions, such as the prioritization of NEV deployment in high-emission zones and timing traffic management measures to mitigate peak pollution periods. The scenarios for NEV penetration are also forward-thinking, modeling emissions reductions with increased adoption of electric vehicles (EVs) and revealing reductions of up to 80% in certain pollutants in high-penetration scenarios. Such modeling highlights the potential benefits of decarbonizing transportation sectors within heavily trafficked urban centers.

This work is a valuable addition to the field of atmospheric chemistry and urban environmental management, providing a replicable model for other cities aiming to control air pollution at a fine scale. Future research could further enhance this study by incorporating low-cost sensor data or machine-learning-based gap-filling to expand coverage and reduce dependency on fixed monitoring networks. The

paper underscores the transformative potential of AI and big data in developing comprehensive, dynamically updated emissions inventories that support air quality improvements and health outcomes.

35 **Response to General Comments:** We sincerely thank Dr. Singh for the thoughtful evaluation and constructive feedback on our study. We appreciate the recognition of the methodological innovations and practical implications of integrating real-time traffic data with big data approaches for high-resolution emission mapping. The suggestions on enhancing interdisciplinary applications and future research directions have been carefully addressed in the revised manuscript to strengthen the scientific rigor and 40 policy relevance.

Detailed point-by-point responses to all comments are provided below. The comments are in black, while our responses and changes in the manuscript are in blue and in blue italic, respectively.

**Specific Comments:**

**Comment #1:** The current reliance on fixed traffic cameras could be complemented by integrating data 45 from mobile low-cost sensors on public vehicles (e.g., buses or taxis) and citizen monitoring devices. This would improve coverage, especially in areas with fewer monitoring points, and provide higher temporal resolution. High-resolution satellite imagery or drone footage could supplement ground-level data, helping to capture emissions near intersections, construction zones, or other areas prone to congestion where cameras may have limited views. Authors of https://arxiv.org/abs/2410.19773 have 50 shown some success in this direction.

**Response to Comment #1:** We sincerely appreciate Dr. Singh's constructive suggestion to integrate mobile low-cost sensors and citizen monitoring devices to further enhance spatial coverage and temporal resolution. Our current study focuses on utilizing existing fixed traffic camera networks and spatial interpolation techniques (e.g., Gaussian smoothing and nearest-neighbor methods). We fully agree the 55 potential advantages of emerging techniques such as high-resolution satellite imagery and drone footage (Ghosal et al., 2024). The proposed methods are promising extensions for capturing hyperlocal emission variability in future research and will be of great benefit to future improvements of this study.

In this work, high spatial resolution of 50 m × 50 m and high temporal resolution of 1 hour were achieved based on the traffic monitoring points in Jinan (supported by more than 3,000 cameras) and interpolation 60 methods, which can basically meet the resolution requirements for street-scale emission mapping. This design effectively addressed spatial gaps between monitoring points while balancing computational feasibility. Nevertheless, we agree that mobile sensors or drone-based data could provide granular insights into localized hotspots (e.g., intersections or construction zones) and will prioritize these approaches in subsequent studies, contingent on data accessibility and collaboration with municipal stakeholders.

**References:**

Ghosal, S., Singh, M., Ghude, S., Kamath, H., SB, V., Wasekar, S., Mahajan, A., Dashtian, H., Yang, Z.-L., Young, M., and Niyogi, D.: Developing Gridded Emission Inventory from High-Resolution Satellite Object Detection for Improved Air Quality Forecasts, arXiv [preprint], arXiv: 2410.19773, 14 October 2024.

**Comment #2:** Traditional emission factors could be replaced or augmented by machine learning models that dynamically predict emissions based on vehicle type, traffic density, and weather conditions. This could improve accuracy, especially for rapidly changing urban traffic conditions. Adding localized, real-time meteorological data (e.g., wind speed, temperature) from additional sources such as local weather stations or remote sensors would improve the emission model by accounting for variations in atmospheric dispersion conditions.

**Response to Comment #2:** We sincerely thank Dr. Singh for highlighting the potential of machine learning (ML) models to enhance dynamic emission predictions. We fully agree that integrating ML-based emission factors (e.g., incorporating real-time traffic density, vehicle types, and hyperlocal weather conditions) is a good idea, as it could improve model adaptability to rapidly changing urban scenarios. However, we acknowledge that transitioning to ML-driven emission factors requires extensive training data (e.g., vehicle-specific telemetry, high-frequency meteorological measurements) and robust validation frameworks, which are currently beyond the scope of this work.

In the current study, we adopted conventional emission factors to align with the widely recognized methodologies in Technical Guide for Compilation of Atmospheric Pollutants Emission Inventory from Road Motor Vehicles (Trial) (MEE, 2014). This approach ensures comparability with prior studies and facilitates policy applications. In addition, with consideration of emission variations caused by local conditions, localized corrections (e.g., environmental correction and traffic condition correction) were applied in this study for adjusting emission factors.

To address this limitation, we have revised the Discussion section to explicitly propose ML-based emission factor optimization as a critical future direction. Specifically, we emphasize the need for collaborations with transportation agencies to access granular vehicle activity data. We added the following to the revised manuscript.

**Lines 555-557:** *"Machine learning (ML) can further enhance the framework by dynamically optimizing emission factors based on traffic patterns, vehicle types, and meteorological conditions, thereby achieving real-time traffic data processing and dynamic updates of vehicle emission inventory."*

**References:**

MEE (Ministry of Ecology and Environment of the People's Republic of China): Technical guidelines for compiling atmospheric pollutant emission inventory of road motor vehicles (Trial), available at: https://www.mee.gov.cn/gkml/hbb/bgg/201501/W020150107594587831090.pdf (last access: 17 May 2024), 2014 (in Chinese).

**Comment #3:** Expanding vehicle classifications (e.g., electric, plug-in hybrid, fuel cell, diesel) would allow for finer distinctions in emissions and support more precise electrification scenarios. Integrating land use data (e.g., residential, industrial, commercial) could reveal patterns in emissions by area type, helping tailor policies for specific zones, such as low-emission zones near schools or hospitals.

**Response to Comment #3:** We sincerely thank Dr. Singh for the valuable suggestions to refine vehicle classifications and integrate land use data. These points are indeed critical for advancing electrification scenarios precision, and making spatially targeted recommendations.

While expanding vehicle categories (e.g., electric, plug-in hybrid, fuel cell) would enhance electrification scenario analysis, the current study is constrained by the fineness of available traffic monitoring data. In Jinan's traffic camera network, specific types of NEVs cannot be distinguished, due to technical and regulatory limitations in real-time identification of specific powertrain technologies. We acknowledged this as a limitation and had provided clarification and specific explanations in Section 2.5. NEVs are mainly classified into battery electric vehicles (BEVs), plug-in hybrid electric vehicles (PHEVs), and fuel cell vehicles (FCVs). Except for PHEVs, which emit pollutants in hybrid mode, all other NEVs produce no pollutants during driving. As to PHEVs, they only account for a relatively small proportion within NEVs and they primarily operate in electric mode in short-distance driving. Therefore, conducting scenario simulation without fine classification of different types of NEVs will not result in a large error. We fully agree that land use patterns can reveal emission hot spots tied to urban functions. We would like to clarify that the available land use data in Jinan categorizes areas primarily into broad classes (e.g., water bodies, vegetation, and built-up areas) due to limitations in publicly accessible geographic datasets. Nevertheless, a rough analysis based on city maps shows that emissions are generally lower in residential areas than in commercial areas, and that emission hot spots are concentrated in central commercial areas, while emission cold spots usually occur in residential areas, as detailed in Fig.9 and Lines 445-449.

---

## Author Comment (AC3)

**Response to Comments from Referee #2**

**Manuscript:** egusphere- 2024-2791

**Title:** High-resolution mapping of on-road vehicle emissions with real-time traffic datasets based on big data

**Authors:** Yujia Wang et al.

**Corresponding author:** Xinfeng Wang (email: xinfengwang@sdu.edu.cn)

**General Comments:** The paper reports a methodology and an analysis of high spatial resolution emissions of road vehicles in Jinan (China) based on real-time traffic data. Some future scenarios are discussed related to penetration of electric vehicles. The topic is interesting and useful for planning future mitigation strategies in urban areas. However, some aspects are not clear and a revision will be beneficial.

**Response to General Comments:** We thank the referee for the valuable feedback and for recognizing the usefulness of our study for planning future mitigation strategies in urban areas. We also appreciate the constructive comments and suggestions for the improvements and clarifications. Based on the comments and suggestions, the original manuscript is accordingly revised. Especially, we have made careful thinking and revisions to make the descriptions more clearly, so as to improve the rigor and scientific nature of the discussion.

Listed below are the detailed point-by-point responses and resulting edits to all the comments. The review comments are in black, while our responses and changes in the manuscript are in blue and in blue italic, respectively.

**Specific Comments:**

**Comment #1:** Lines 35-50. In this discussion, I suggest to mention the methodologies used to determine emissions of vehicles in real operating conditions such as eddy-covariance approach, see foe example, Conte et al. (Environmental Pollution 251, 830-838, 2019).

**Response to Comment #1:** We thank the referee for making this valuable suggestion. We agree that it is necessary to mention the methodology used to determine emissions of vehicles in real operating conditions, as they provide accurate and realistic vehicle emission factors. Vehicle emission factors are one of the basic data for the development of on-road vehicle emission inventories. As suggested, we have added a description of the methods used to determine emission factors of vehicles in real operating conditions in the revised manuscript.

**Lines 35–42:** "*Emission factors, defined as the number (or mass) of pollutants emitted per unit of activity, are one of the basic data for the development of emission inventories. Several methods for determining on-road vehicle emission factors have been used in real-world conditions, such as street canyon and road tunnel studies (Brimblecombe et al., 2015; Zhang et al., 2015), remote sensing (Davison et al., 2020), and on-board measurements (Jaikumar et al., 2017). Other methods involve the estimation of air dispersion by tracking a trace gas that can be added or emitted by traffic (Belalcazar et al., 2010) and the use of the eddy-covariance (EC) technique to quantify emissions based on atmospheric turbulence (Conte and Contini, 2019). Although previous measurements have yielded convincing emission factors for local vehicles, some limitations in the development of emission inventories have not yet been adequately addressed.*"

**References:**

Belalcazar, L. C., Clappier, A., Blond, N., Flassak, T., and Eichhorn, J.: An evaluation of the estimation of road traffic emission factors from tracer studies, Atmos. Environ., 44, 3814–3822, https://doi.org/10.1016/j.atmosenv.2010.06.038, 2010.

Brimblecombe, P., Townsend, T., Lau, C. F., Rakowska, A., Chan, T. L., Močnik, G., and Ning, Z.: Through-tunnel estimates of vehicle fleet emission factors, Atmos. Environ., 123, 180–189, https://doi.org/10.1016/j.atmosenv.2015.10.086, 2015.

Conte, M. and Contini, D.: Size-resolved particle emission factors of vehicular traffic derived from urban eddy covariance measurements, Environ. Pollut., 251, 830–838, https://doi.org/10.1016/j.envpol.2019.05.029, 2019.

Davison, J., Bernard, Y., Borken-Kleefeld, J., Farren, N. J., Hausberger, S., Sjödin, Å., Tate, J. E., Vaughan, A. R., and Carslaw, D. C.: Distance-based emission factors from vehicle emission remote sensing measurements, Sci. Total Environ., 739, 139688, https://doi.org/10.1016/j.scitotenv.2020.139688, 2020.

Jaikumar, R., Shiva Nagendra, S. M., and Sivanandan, R.: Modal analysis of real-time, real world vehicular exhaust emissions under heterogeneous traffic conditions, Transp. Res. Part Transp. Environ., 54, 397–409, https://doi.org/10.1016/j.trd.2017.06.015, 2017.

Zhang, Y., Wang, X., Li, G., Yang, W., Huang, Z., Zhang, Z., Huang, X., Deng, W., Liu, T., Huang, Z., and Zhang, Z.: Emission factors of fine particles, carbonaceous aerosols and traces gases from road vehicles: Recent tests in an urban tunnel in the Pearl River Delta, China, Atmos. Environ., 122, 876–884, https://doi.org/10.1016/j.atmosenv.2015.08.024, 2015.

**Comment #2:** Section 2.2. It is not clear how the meteo data of ERA are used in this approach even considering that the spatial resolution is much lower than that of calculated emissions. In addition, what are the emission factors used?

**Response to Comment #2:** We thank the referee for the good question and regret for the insufficient discussion. Regarding the use of ERA meteorological data, although the spatial resolution of ERA data is much lower than that of the calculated emissions, its high temporal resolution (hourly) and long-term completeness and reliability provide an ideal meteorological background for this study. To address the spatial resolution mismatch, we aligned the meteorological data with the emissions data by assigning meteorological values to the nearest geographic coordinates. Although this approach may result in multiple traffic monitoring points sharing the same meteorological data, it ensures that each point has complete meteorological information for subsequent calculations. It is worth noting that ERA meteorological data were primarily used for environmental correction of emission factors. In this process, we only need to use the numerical ranges of local temperature and humidity and assign specific correction coefficients accordingly, as detailed in the Response to Comment #4. Therefore, the spatial resolution of meteorological data has minimal impact on the study results. Even if the same meteorological data are applied to multiple monitoring points, generally accurate environmental correction coefficients can be obtained, ensuring the reliability of the analysis.

The emission factors (EFs) used in this study were the products of the comprehensive baseline emission factors and the local correction coefficients (see Eq. 1). All comprehensive baseline emission factors (BEF) and correction coefficients were adopted from the Technical Guide for Compilation of Atmospheric Pollutants Emission Inventory from Road Motor Vehicles (Trial) (MEE, 2014).

$$EF_{c,j} = BEF_{c,j} \times \varphi \times \gamma \times \lambda \times \theta, \tag{Eq. 1}$$

Where, $BEF_{c,j}$ is the comprehensive baseline emission factor of pollutant $j$ for vehicle category $c$, in unit of grams per kilometer (g km$^{-1}$). The $BEF_{c,j}$ was calculated based on various vehicle types in China, considering the average cumulative driving mileage and the following scenarios: typical urban driving conditions (30 km/h), meteorological conditions (temperature of 15°C, relative humidity of 50%), fuel quality (gasoline and diesel sulfur content of 50 ppm and 350 ppm, respectively, with no ethanol blending in gasoline), and load factor (50% load factor for diesel vehicles under typical operating conditions). $\varphi$, $\gamma$, $\lambda$, and $\theta$ are the dimensionless environmental correction coefficient, traffic condition correction coefficient, deterioration correction coefficient, and vehicle usage conditions correction coefficient (*e.g.*, the load of diesel vehicles), respectively. The meteorological data obtained from ERA5 were used to

determine environmental correction coefficient $\varphi$. For greater clarity of description, we added the following to the revised manuscript.

**Lines 134-135:** *"Meteorological data were used for environmental corrections of emission factors. In this process, we determined the local temperature and humidity ranges rather than the exact values, and assigned specific correction coefficients accordingly."*

**References:**

MEE (Ministry of Ecology and Environment of the People's Republic of China): Technical guidelines for compiling atmospheric pollutant emission inventory of road motor vehicles (Trial), available at: https://www.mee.gov.cn/gkml/hbb/bgg/201501/W020150107594587831090.pdf (last access: 17 May 2024), 2014 (in Chinese).

**Comment #3:** Another aspect, section 2.3, not clear if the electrical vehicles. It is assumed that these have zero emissions but this is not true in general. A large fraction (more than 50%) of emissions in non-electric vehicles is due to non-exhaust and this will be present, and likely increased, in electric vehicles. Please justify this choice and if it was not considered the non-exhaust certainly the reduction to new vehicles penetration is largely overestimated.

**Response to Comment #3:** We appreciate the referee's important comments. Non-exhaust emission sources (e.g., road dust, brake wear, and tire wear) account for a significant proportion of traffic emissions. Additionally, electrical vehicles are typically heavier than conventional vehicles, which may lead to higher non-exhaust emissions (Timmers and Achten, 2016; Liu et al., 2021). In this study, the scenario simulation of fleet electrification was conducted based on the exhaust emission inventory, focusing primarily on the differences in exhaust emissions between internal combustion engine vehicles and new energy vehicles. As a result, our calculations and analyses regarding the impact of increased NEV penetration on emissions were based solely on exhaust emission sources. It is undeniable that if the non-exhaust emissions were also considered, the emission reduction benefits of increased NEV penetration would be lower than our current analysis results, particularly in PM emissions. With the upgraded limits of tailpipe exhaust and the increasing share of electric vehicles, non-exhaust emissions are becoming increasingly prominent (Zhang et al., 2020). Therefore, future research needs to comprehensively consider the contributions of both exhaust and non-exhaust emissions to provide a more thorough evaluation of the impact of NEV adoption on air quality improvement. To avoid potential misunderstandings caused by insufficient discussion, we added the following sentences to the revised manuscript,

**Lines 497-504:** "*Note that when evaluating the impact of NEV penetration on on-road vehicle emissions, our calculations and analyses were based merely on exhaust emissions, focusing on the differences in exhaust emissions between ICEVs and NEVs. However, a substantial portion of PM emissions are contributed by non-exhaust sources (e.g., road dust, brake wear, and tire wear) (Zhang et al., 2020). Additionally, NEVs are typically heavier than conventional vehicles, which may lead to higher non-exhaust emissions (Timmers and Achten, 2016; Liu et al., 2021). Therefore, if the contribution of non-exhaust emissions was also taken into account, the estimated benefits of increased NEV penetration on reducing PM$_{2.5}$ emissions would be lower than suggested by our current analysis.*"

**References:**

Liu, Y., Chen, H., Gao, J., Li, Y., Dave, K., Chen, J., Federici, M., and Perricone, G.: Comparative analysis of non-exhaust airborne particles from electric and internal combustion engine vehicles, J. Hazard. Mater., 420, 126626, https://doi.org/10.1016/j.jhazmat.2021.126626, 2021.

Timmers, V. R. J. H. and Achten, P. A. J.: Non-exhaust PM emissions from electric vehicles, Atmos. Environ., 134, 10–17, https://doi.org/10.1016/j.atmosenv.2016.03.017, 2016.

Zhang, J., Peng, J., Song, C., Ma, C., Men, Z., Wu, J., Wu, L., Wang, T., Zhang, X., Tao, S., Gao, S., Hopke, P. K., and Mao, H.: Vehicular non-exhaust particulate emissions in Chinese megacities: Source profiles, real-world emission factors, and inventories, Environ. Pollut., 266, 115268, https://doi.org/10.1016/j.envpol.2020.115268, 2020.

**Comment #4:** Lines 159-161. What kind of correction and with what spatial resolution? The same in line 339.

**Response to Comment #4:** We regret for the misleading. The correction referred to the standard scenario based on baseline emission factors (BEF) measurement, following the methods and values provided in the guidelines. Specifically, environmental correction involved distributing grid meteorological data with a resolution of 0.25° to traffic monitoring points to determine the temperature and humidity ranges at these points. The corresponding correction coefficients, as listed in Tables S3–S6, were then matched. Traffic condition correction was based on road congestion data provided by Gaode Maps (https://www.amap.com/), estimating the speed range and matching the corresponding correction coefficients, as listed in Tables S7–S8. The supplement to the manuscript contains additional data and explanations. We added the following words and tables to the revised manuscript and supplement,

**Lines 169-171:** "*In addition, environmental correction was conducted mainly based on temperature and humidity which varied largely from season to season, as detailed in Tables S3–S6. As to the traffic*

*condition correction, coefficients were determined based on the average vehicle speed intervals (see Tables S7–S8)."*

**Supplement lines 47-59:**

**Table S3.** *Temperature correction coefficients for gasoline vehicles.*

| Pollutants | Low temperature (<10°C) | High temperature (>25°C) |
|---|---|---|
| CO | 1.36 | 1.23 |
| HC | 1.47 | 1.08 |
| NO$_x$ | 1.15 | 1.31 |

**Table S4.** *Temperature correction coefficients for diesel vehicles.*

| Pollutants | Vehicle classification | Low temperature (<10°C) | High temperature (>25°C) |
|---|---|---|---|
| CO | LDPV | 1.00 | 1.33 |
| | LDT | 1.00 | 1.33 |
| | MDPV, HDPV, Bus, MDT, HDT | 1.00 | 1.30 |
| HC | LDPV | 1.00 | 1.07 |
| | LDT | 1.00 | 1.06 |
| | MDPV, HDPV, Bus, MDT, HDT | 1.00 | 1.06 |
| NO$_x$ | LDPV | 1.06 | 1.17 |
| | LDT | 1.05 | 1.17 |
| | MDPV, HDPV, Bus, MDT, HDT | 1.06 | 1.15 |
| PM$_{2.5}$ | LDPV | 1.87 | 0.68 |
| | LDT | 1.27 | 0.90 |
| | MDPV, HDPV, Bus, MDT, HDT | 1.70 | 0.74 |

**Table S5.** *Humidity correction coefficients for gasoline vehicles.*

| Pollutants | Temperature | Low humidity (<50%) | High humidity (>50%) |
|---|---|---|---|

| | >24°C | 0.97 | 1.04 |
|---|---|---|---|
| CO | >24°C | 0.97 | 1.04 |
| HC | >24°C | 0.99 | 1.01 |
| NO$_x$ | >24°C | 1.13 | 0.87 |
| | <24°C | 1.06 | 0.92 |

| Pollutant | Temperature | Low | High |
|---|---|---|---|
| CO | >24°C | 0.97 | 1.04 |
| HC | >24°C | 0.99 | 1.01 |
| NO$_x$ | >24°C | 1.13 | 0.87 |
| | <24°C | 1.06 | 0.92 |

**Table S6.** *Humidity correction coefficients for gasoline vehicles.*

| Pollutants | Temperature | Low humidity (<50%) | High humidity (>50%) |
|---|---|---|---|
| NO$_x$ | >24°C | 1.12 | 0.88 |
| | <24°C | 1.04 | 0.94 |

**Table S7.** *Correction coefficients for average traveling speed for diesel vehicles.*

| Pollutants | Speed intervals (km/h) | | | | |
|---|---|---|---|---|---|
| | <20 | 20-30 | 30-40 | 40-80 | >80 |
| CO | 1.69 | 1.26 | 0.79 | 0.39 | 0.62 |
| HC | 1.68 | 1.25 | 0.78 | 0.32 | 0.59 |
| NO$_x$ | 1.38 | 1.13 | 0.90 | 0.86 | 0.96 |
| PM$_{2.5}$ | 1.68 | 1.25 | 0.78 | 0.32 | 0.59 |

**Table S8.** *Correction coefficients for average traveling speed for diesel vehicles.*

| Pollutants | Speed intervals (km/h) | | | | |
|---|---|---|---|---|---|
| | <20 | 20-30 | 30-40 | 40-80 | >80 |
| CO | 1.29 | 1.10 | 0.93 | 0.70 | 0.61 |
| HC | 1.38 | 1.12 | 0.91 | 0.64 | 0.48 |
| NO$_x$ | 1.39 | 1.12 | 0.91 | 0.60 | 0.28 |
| PM$_{2.5}$ | 1.36 | 1.12 | 0.91 | 0.65 | 0.48 |

*Notes: Buses are usually corrected for <20 km/h.*

**Comment #5:** Line 529. Better to say future scenarios, otherwise the sentence is strange.

**Response to Comment #5:** This sentence has been modified as follows in the revised manuscript.

**Lines 547-548:** *"These results not only demonstrate the potential of fleet electrification in emission reduction, but also provide a scientific basis for formulating more precise emission reduction strategies."*